

# Daytime HONO, NO₂ and aerosol distributions from MAX-DOAS observations in Melbourne

Robert G. Ryan[1,2], Steve Rhodes[3], Matthew Tully[3], Stephen Wilson[4], Nicholas Jones[4], Udo Frieß[5], and Robyn Schofield[1,2]

[1]School of Earth Sciences, University of Melbourne, Melbourne, Australia
[2]ARC Centre of Excellence for Climate System Science, Sydney, Australia
[3]Bureau of Meteorology, Melbourne, Australia
[4]School of Chemistry, University of Wollongong, Australia
[5]Institute for Environmental Physics, University of Heidelberg, Germany

**Correspondence:** Robert G. Ryan (rgryan@student.unimelb.edu.au)

**Abstract.** Nitrogen oxides produced by high temperature combustion are prevalent in urban environments and toxic, contributing to a significant health burden. The chemistry of nitrogen oxides such as NO₂ and HONO in pollution are important for hydroxyl radical production and overall oxidative capacity in urban environments, however current mechanisms cannot explain high daytime levels of HONO observed in many urban and rural locations around the world. Here we present HONO, NO₂ and

aerosol extinction vertical distributions retrieved from MAX-DOAS measurements in suburban Melbourne, which are the first MAX-DOAS results from Australia. Using the optimal estimation algorithm HEIPRO we show that vertical profiles for NO₂ and HONO can be calculated with low dependence on the retrieval forward model and a priori parameters, despite a lack of independent co-located aerosol or trace gas measurements. Between December 2016 and April 2017 average peak NO₂ values of $8 \pm 2$ ppb indicated moderate traffic pollution levels, and high daytime peak values of HONO were frequently detected, av-

eraging $220 \pm 30$ ppt in the middle of the day. HONO levels measured in Melbourne were typically lower than those recorded in the morning in other places around the world, indicating minimal overnight accumulation, but peaked in the middle of the day to be commensurate with midday concentrations in locations with much higher NO₂ pollution. Regular midday peaks in the diurnal cycle of HONO surface concentrations have only previously been reported in rural locations. The HONO measured represents an OH radical source in the middle of the day in Melbourne up to ten times stronger than from ozone photolysis. The

dependence of the high HONO levels on time since rainfall, combined with the observed diurnal and vertical profiles, provide evidence for a strong photo-activated and ground-based daytime HONO source.

*Copyright statement.* TEXT





# 1 Introduction

The World Health Organisation indicates that ambient air pollution exposure presents the largest environmental risk to human health, with as many as one in nine deaths attributable to poor air quality (WHO, 2016). In Australia, with isolated but highly urbanised population centres, the economic burden attributed to health care as a result of air pollution is estimated to be as high as \$11 million per year (DEE, 2018). This makes understanding the oxidation chemistry underpinning urban pollution processes, and particularly closing the budget of chemical oxidants in urban areas, a priority for atmospheric scientists. The oxidative capacity of the atmosphere can be defined by its main oxidant, the hydroxyl radical (OH), yet relative contributions of different processes to local and regional OH budgets remains uncertain. Ozone photolysis through R1 and R2 is often considered to be the primary pathway for atmospheric OH formation in the boundary layer:

$$O_3 + h\nu \quad \rightarrow \quad O(^1D) + O_2 \tag{R1}$$
$$O(^1D) + H_2O \quad \rightarrow \quad 2OH \tag{R2}$$

Several studies have identified higher daytime nitrous acid (HONO) levels in urban areas than can be expected from the known mechanisms, indicating an unknown daytime HONO source (Lee et al., 2016; Acker and Möller, 2007; Kleffmann, 2007; Wong et al., 2012; Huang et al., 2017; Neuman et al., 2016). Given that HONO photolysis through R3 is a strong OH production source, elevated daytime HONO levels can increase the local tropospheric oxidative capacity:

$$HONO + h\nu(300 - 405nm) \rightarrow NO + OH \tag{R3}$$

Major known HONO sources include direct emission from combustion engines, the daytime homogeneous reaction R4, and the heterogeneous reaction R5 occurring on wet surfaces which is believed be the main nighttime HONO source reaction.

$$NO + OH + M \quad \rightarrow \quad HONO + M \tag{R4}$$
$$2NO_2 + H_2O_{(ads)} \quad \rightarrow \quad HONO + HNO_{3(ads)} \tag{R5}$$

HONO sinks include dry deposition, photolysis to produce OH (R3) and reaction with OH (R6).

$$HONO + OH \rightarrow H_2O + NO_2 \tag{R6}$$

Since R6 is very slow (rate constant $1.8 \times 10^{-11} s^{-1}$) compared to R3 (photolysis rate $J(HONO) \approx 3 \times 10^{-5} s^{-1}$ around midday) (Sander et al., 2006), photolysis is the dominant daytime sink process. Consequently, HONO accumulating overnight rapidly photolyses in the early morning, and HONO concentrations are expected to decrease with increasing UV radiation. While observing daytime HONO decreases from an early morning maximum, many studies have still observed higher than





expected daytime HONO concentrations, with a missing source component peaking around the middle of the day (Li et al., 2012; Qin et al., 2009; Lee et al., 2016; Hendrick et al., 2014; Pinto et al., 2014). Maximum diurnal HONO mixing ratios have been reported during the daytime in Cyprus (Meusel et al., 2016) and at a rural site in Germany (Acker et al., 2006). Studies have suggested the missing source may be related to heterogeneous chemistry involving water, aerosols, ground surfaces, or soil-based emissions. To assess the feasibility of these potential HONO sources, reliable vertical gradient measurements of HONO and its precursors, including nitrogen dioxide ($NO_2$) are required. To address this need, in recent times, passive monitoring techniques such as multi-axis differential optical absorption spectroscopy (MAX-DOAS) have risen to prominence.

The MAX-DOAS technique relies on measurements of scattered sunlight in the ultra violet and visible wavelengths (UV-vis), at several different viewing angles, facilitating the retrieval of vertical information on tropospheric aerosol extinction and trace gas concentration (Platt and Stutz, 2008; Hönninger et al., 2004). Variations in light pathlength due to aerosol scattering are inferred from absorption measurements of the oxygen dimer collision complex $O_4$, which has a well defined relationship with atmospheric pressure (Wagner et al., 2004; Frieß et al., 2006). The retrieved aerosol information can then be used as input to estimate vertical concentration profiles for UV-vis absorbing trace gases including $NO_2$, HONO, formaldehyde, glyoxal and bromine monoxide e.g. (Vlemmix et al., 2015; Hendrick et al., 2014; Schreier et al., 2016; Jin et al., 2016).

$NO_2$ has been well studied using MAX-DOAS in many locations around the world due to its strong UV absorbance and its ubiquity as an urban pollutant, e.g. (Ma et al., 2013; Vlemmix et al., 2015; Kanaya et al., 2014; Wagner et al., 2011; Ortega et al., 2015). In contrast nitrous acid has more commonly been studied using active, long-path DOAS due to its lower concentration and weaker absorbance (Platt and Perner, 1983; Kleffmann et al., 2006; Stutz et al., 2010). Recently attention has turned to studying nitrous acid using MAX-DOAS, as shown in a HONO slant column intercomparison during the MADCAT campaign in Mainz, Germany (Wang et al., 2017) and in HONO and $NO_2$ profile retrievals in Beijing (Hendrick et al., 2014). The findings of Hendrick et al. (2014) were in line with previous published HONO data from long path DOAS and in situ measurements, showing higher than expected HONO concentrations and indicating an unaccounted source of daytime HONO e.g. (Lee et al., 2016; Acker and Möller, 2007; Kleffmann, 2007; Wong et al., 2012; Huang et al., 2017; Neuman et al., 2016). Considerable efforts have been made to determine the mechanism of the missing source(s), and the importance of HONO as a tropospheric radical source through vertical gradient measurements using long-path DOAS (Stutz et al., 2010; Wong et al., 2012; Young et al., 2012) aeroplanes (Neuman et al., 2016), zeppelins (Li et al., 2014) and towers (Kleffmann et al., 2003; VandenBoer et al., 2013).

Compared to these expensive, short term campaign platforms, MAX-DOAS measurements of HONO vertical profiles as demonstrated in Hendrick et al. (2014) have the significant advantages of simple autonomous instrumentation, being cheap to run and able to be deployed in any environment for long term monitoring programs. However, the MAX-DOAS method has some drawbacks which include complicated, multi-step data processing and limited information content from which to draw vertical profile information. Furthermore, uncertainties in forward model and profile a priori parameters can lead to measurement errors which must be quantified for confidence in the retrieval. While in some locations this is facilitated by co-located aerosol and trace gas measurements, in the absence of such external data, poorly constrained parameters can introduce large errors (Ortega et al., 2016; Wagner et al., 2011).



While MAX-DOAS has been deployed across much of the northern hemisphere, there are few reports of DOAS and HONO observations from the Southern Hemisphere. In this paper measurements are presented from Melbourne which are, to the best of our knowledge, the first MAX-DOAS results published from Australia. Melbourne, the capital of Victoria, is the second largest city in Australia, with over 4.8 million people, accounting for 19% of the national population. The Victorian Environmental
Protection Agency (EPA) has been monitoring air quality, including particulate pollution and $NO_2$ levels at several different sites around the Melbourne metropolitan area since 1979 (see map in fig. 1). In that time average annual $NO_2$ levels have decreased from 13 ppb to below 10 ppb, despite significant population and vehicle number increases (EPA, 2013), attributed to improved vehicle pollution reduction technology and fuel efficiency. With the Australian National Environment Protection Measure Standards annual average $NO_2$ concentration at 30 ppb, these figures indicate that Melbourne has good air quality
as far as nitrogen oxides are concerned. On the one hand, this provides an ideal opportunity to study the oxidation budget of HONO and OH in a low to moderately polluted urban environment. On the other hand, addressing the paucity of air quality data in Melbourne is relevant given that epidemiological studies have demonstrated correlations between particulates and $NO_2$ pollution on overall mortality (Simpson et al., 2000, 2005) and cardiovascular disease (Barnett et al., 2006) in Melbourne. The results presented in this paper demonstrate the ability of MAX-DOAS measurements to address the air quality data deficit in
Melbourne, locally, and the Southern Hemisphere more broadly, and contribute to an improved understanding of how HONO impacts the budget of tropospheric oxidants.

## 2 Measurement and profile retrieval details

### 2.1 Measurement site and MAX-DOAS instrumentation

The Australian Bureau of Meteorology (BOM) has operated a MAX-DOAS instrument in Broadmeadows, a northern suburb
of Melbourne, since August 2016. Results are presented in this work from December 2016 to April 2017. The instrument is mounted on a laboratory roof looking in a south-westerly direction over one of Melbourne's main arterial motorways (the Western Ring Road) and, further to the south, the northern suburbs and central city as shown in fig. 1. Being close to the Western Ring Road, the MAX-DOAS is ideally placed to measure the resulting traffic pollution plumes. Being on the northern fringes of the Melbourne metropolitan area, the instrument is well placed to study the interaction of rural and urban air masses
given the appropriate prevailing meteorology.

The MAX-DOAS used in this work was a commercial 1-D instrument manufactured by the German company Environmental Measurement Systems (Envimes). The instrument consists of a scanner telescope box, looking towards a fixed compass direction of 208 degrees, connected by fibre optic and data cables to spectrometer and computer units inside the laboratory. The spectrometer unit contained temperature stablized 75 mm Avantes Spectrometers for UV (295-450 nm, 0.6 nm resolution)
and and visible (430-565 nm, 0.6 nm resolution) regions. The UV detector was a Hamamatsu backthinned detector with Schott BG3 fitler and 2048 pixel channels while the visible detector was a Sony 2048L also with 2048 channels. The telescope unit contained a rotating prism and inclinometer facilitating active elevation control with quoted elevation angle accuracy < 0.1°. Similar commercial grade Envimes MAX-DOAS instruments have demonstrated good performance at the MAD-CAT inter-



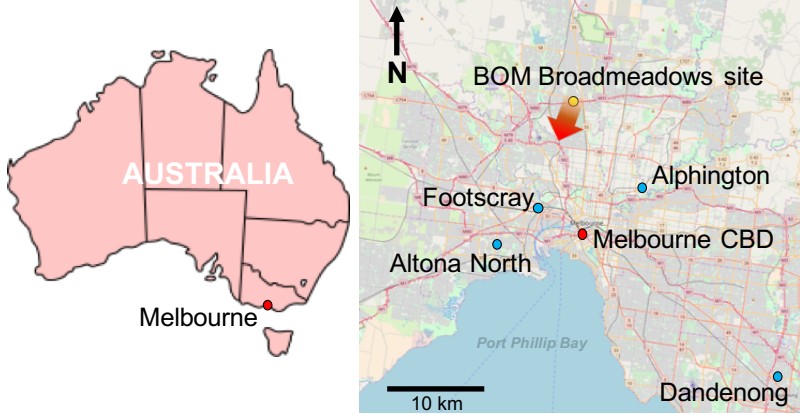

**Figure 1.** Left: map of Australia showing the location of Melbourne. Right: Map of the Melbourne metropolitan area showing the location of the Broadmeadows measurement site (yellow marker) in the northern suburbs. The red arrow shows the south-south-west viewing direction of the MAX-DOAS instrument. The blue markers indicate the four Victorian EPA $NO_2$ measurement sites which are referred to in the results section.

comparison campaign in Mainz, Germany (Lampel et al., 2015; Wang et al., 2017). Wavelength calibrations were carried out using an external mercury lamp, and spectra were corrected for detector non-linearity, dark current and spectral offset using laboratory measurements. The measurement sequence was controlled using the MS-DOAS software custom-designed by EnviMes, and consisted of a set of elevation angle scans at 90°, 30°, 20°, 10°, 5°, 3° and 2° which took approximately 12 minutes

to complete.

## 2.2  DOAS fitting

The DOAS technique allows the the Beer-Lambert Law to be applied in an atmospheric context, with 'low frequency' attenuation components of the scattered light spectra (such as Rayleigh and Mie scattering) being separated from the 'high frequency' trace gas absorptions (Platt and Stutz, 2008). The low frequency component is approximated using a polynomial as shown in

fig. 2(a). The high frequency component is fitted with the relevant trace gas absorption cross sections using a least squares fitting algorithm. The resulting value is the differential slant column density, which is the light-path integrated trace gas concentration, relative to a reference spectrum. In the case of MAX-DOAS measurements the reference spectrum is typically the 90° scan recorded as part of a set of measurements at different elevation angles. This reference method cancels out most of the stratospheric influence, allowing the retrieval of tropospheric specific information (Platt and Stutz, 2008).

DOAS analysis was carried out using the QDOAS software developed at BIRA-IASB (http://uv-vis.aeronomie.be/software/QDOAS/). Cross sections used (see table 1) were convolved with the instrumental slit function, measured using mercury emission lines, in QDOAS, and the UV wavelength range 338-370 nm for $NO_2$ and $O_4$ were based on recommended settings from the CINDI intercomparison campaign (Roscoe et al., 2010), with the inclusion of the HONO cross section and a temperature dependent





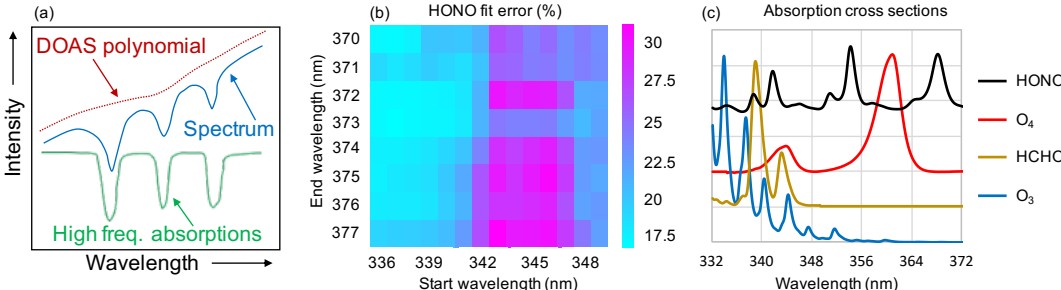

**Figure 2.** (a) Schematic of the DOAS principle, (b), HONO sensitivity study results using the retrieval interval mapping technique and (c) HONO, $O_4$, $O_3$ and HCHO cross sections plotted between 332 and 372 nm, showing the cross section overlap in the HONO fitting interval.

**Table 1.** Details of the DOAS settings in this work

| Species | Cross section |
| --- | --- |
| $O_3$ (223 and 243 K) | Serdyuchenko et al. (2014) |
| $NO_2$ (220 and 298 K) | Vandaele et al. (1998) |
| $O_4$ (293 K) | Thalman and Volkamer (2013) |
| HCHO (297 K) | Meller and Moortgat (2000) |
| BrO (223 K) | Fleischmann et al. (2004) |
| HONO | Stutz et al. (2000) |
| Ring effect (293 K and 250 K) | Grainger and Ring (1962) |
| DOAS polynomial | 5th order |
| Offset term | 1st order |

Ring term as in Volkamer et al. (2015); Lampel et al. (2017). A sensitivity study to optimise the HONO fitting wavelength range was performed using the retrieval interval mapping technique of Vogel et al. (2013) (fig. 2(b)) which showed that the smallest HONO fit percentage errors correspond to a wavelength range starting below 340 nm . When compared to the plot showing all cross sections fitted in the wavelength ranges in fig. 2(c), these results show that the HONO fit error is improved by

5 including all three of the largest HONO cross section peaks, including the 341 nm peak which overlaps strongly with formalde-hyde (HCHO) and $O_4$ peaks. This is in line with Wang et al. (2017) and Hendrick et al. (2014), although the fitting window 339-372 nm chosen here for HONO is shorter than in these papers in order to minimise the overall residual RMS of the fits, while maintaining low HONO fit errors.

Successful DOAS retrievals for $O_4$, $NO_2$ and HONO are demonstrated in fig. 3. In the 339-372 nm fitting window, over the

10 measurement period HONO retrieval errors for solar zenith angles < 80° averaged between 17 % and 25 %, from elevation angles 2° to 30°. These results are not filtered for the influence of clouds. The clear fits shown in fig. 3 and separation between





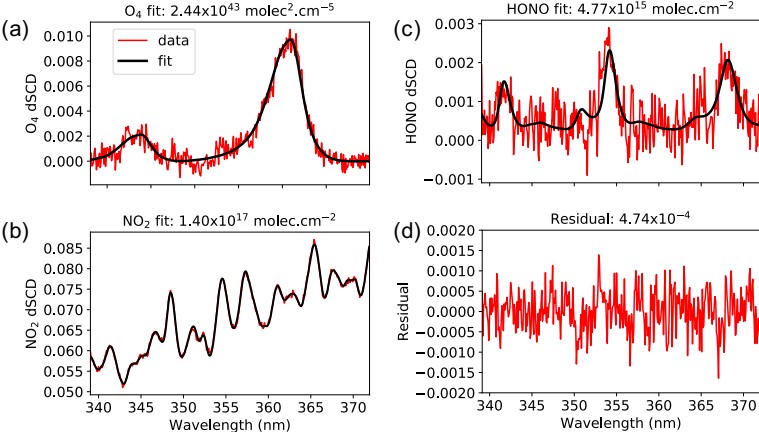

**Figure 3.** Example DOAS fit results for $2°$ elevation angle and $28°$ solar zenith angle on 4th March 2017, showing (a) $O_4$ and (b) $NO_2$ in the 338-370 nm fitting window, with (c) HONO and (d) fit residual in the 339-372 nm fitting range.

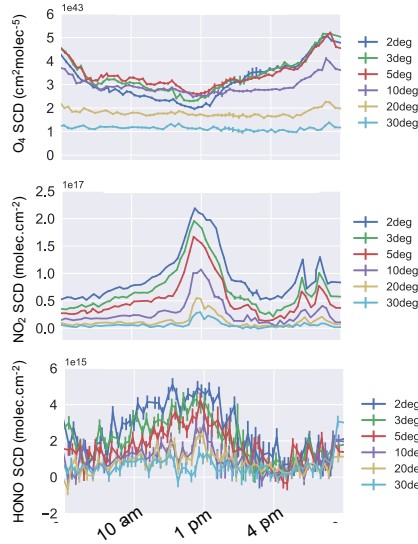

**Figure 4.** Diurnal differential slant column profiles for $O_4$ (top), $NO_2$ (middle) and HONO (bottom) for 4th March 2017.



dSCDs of different elevation angles shown in fig. 4 indicates successful retrieval of HONO slant columns. $NO_2$ and $O_4$ fitting errors were typically of order 2 % throughout the measurement period, and residual RMS values averaged $5.37 \times 10^{-4} \pm 1.13 \times 10^{-4}$ but were typically $< 4.8 \times 10^{-4}$ in the middle of the day as in 3(d), for all elevation angles.

## 2.3  Profile retrieval

A commonly used strategy to retrieve vertical information from MAX-DOAS measurements, involves using a radiative transfer model as a forward model $\mathbf{F}$ to simulate trace gas slant columns. The simulated and measured slant columns are then inverted to calculate a vertical trace gas profile, for example using the optimal estimation method (Rodgers, 1990, 2000; Frieß et al., 2006; Wagner et al., 2004). In order to retrieve trace gas vertical profiles in this way, information on the atmospheric aerosol extinction is needed to constrain the light path. This is determined using by appylying the inversion algorithm to $O_4$ dSCD measurements. The solution for the aerosol profile $\mathbf{x}$ is determined iteratively with input aerosol properties being varied so as to minimise the cost function, given by $\chi^2$, i.e. the difference between the measurement vector $\mathbf{y}$ and the RTM simulations:

$$\chi^2 = (\mathbf{y} - \mathbf{F}(\mathbf{x}))^T \mathbf{S}_\epsilon^{-1} (\mathbf{y} - \mathbf{F}(\mathbf{x})) + (\mathbf{x} - \mathbf{x_a})^T \mathbf{S_a}^{-1} (\mathbf{x} - \mathbf{x}_a) \tag{1}$$

In equation 1, $\mathbf{x}_a$ is a priori information which must be provided to constrain the inversion algorithm because the problem is ill-posed. $\mathbf{S_a}$ and $\mathbf{S}_\epsilon$ represent the error covariance matrices of the a priori and measurement vectors respectively, while $\mathbf{K}$ is the weighting function matrix which describes the sensitivity of the measurement to perturbations in the aerosol profile. The averaging kernel matrix $\mathbf{A} = \frac{\partial \hat{\mathbf{x}}}{\partial \mathbf{x}}$ represents the sensitivity of the retrieved profile $\hat{\mathbf{x}}$ to the true profile $\mathbf{x}$ such that:

$$\hat{\mathbf{x}} = \mathbf{x}_a + \mathbf{A}(\mathbf{x} - \mathbf{x}_a) \tag{2}$$

The information content of a retrieval can be quantified by the degrees of freedom for signal (DOFs), which is the trace of $\mathbf{A}$. The profile retrievals in this work were carried out using the HEIPRO algorithm, as described in Frieß et al. (2006), which uses the radiative transfer code SCIATRAN (Rozanov et al., 2014) as the forward model. HEIPRO also allows for the inclusion of relative intensity measurements in the calculation of aerosol extinction profiles. However, due to the higher sensitivity of intensity measurements to clouds and polarisation, as discussed in Clémer et al. (2010), only $O_4$ dSCDs were used in the retrieval. To ensure agreement between modelled and measured $O_4$ dSCDs, simulations in HEIPRO with different cross sectional scaling factors were carried out as in Wang et al. (2016). It was found that a cross sectional scaling factor of 0.80 on the Hermans et al. (2003) $O_4$ cross section used to model dSCDs in SCIATRAN, consistently brought the measured $O_4$ dSCDs, fitted using the Thalman and Volkamer (2013) $O_4$ cross section, into agreement.

Retrieval of $NO_2$ and HONO profiles in HEIPRO follows the same principles as described for aerosols, where the retrieved aerosol extinction profile for a given scan set of elevation angles is used as input light path information for the trace gas calculations. Aerosol retrievals from $O_4$ were calculated at 360.8 nm, $NO_2$ retrievals at 365.4 nm and HONO retrievals at 354.3 nm. HEIPRO was configured to retrieve profiles over 20 layers from 0.1 to 3.9 km, using elevation angle sets of 90°,



30°, 20°, 10°, 5°, 3° and 2°. For all retrievals a fixed exponentially decreasing a priori profile, characterised by a ground concentration (or ground extinction in the case of aerosols) and a scale height was used to initialise the retrieval. A scale height of 0.6 km was chosen for all retrievals, and the surface parameter (i.e. surface extinction for aerosols, surface concentration for trace gases) varied for each retrieved species. The impact of the choice of these a priori and other forward model parameters

is discussed in further detail in the results section. Construction of the measurement error covariance matrix $\mathbf{S}_\epsilon$ assumed that measurement errors were independent of each other, with diagonal elements equal to the square of the DOAS fit error. The a priori error covariance matrix $\mathbf{S}_a$ was constructed as described by Frieß et al. (2006) with the variance set to 100 % for all altitudes in order to allow for deviations from the a priori in the case of high aerosol optical depths, while minimising opportunities for the algorithm to fit noise. Figure 5 shows some example retrieval results from HEIPRO, for the 7th of March,

a clear sunny day. The profile retrievals are dominated by the layers closest to the ground, as expected given that the averaging kernels show greatest sensitivity in the lowest ≈ 1 km for aerosol extinction and $NO_2$, and the lowest ≈ 500 m for HONO. The high degrees of freedom for signal (DOFs) found for the aerosol and $NO_2$ retrieval gives confidence in the retrieval result. While the DOFs for HONO is lower than for $NO_2$, this result is comparable to the DOFs found for the MAX-DOAS HONO retrieval in Beijing by Hendrick et al. (2014), wherein the difference is attributed to the much greater absorption strength of

$NO_2$ compared to HONO. The combined smoothing and noise errors shown in fig. 5 add up to 10% of the retrieved profile in the lowest 500 m for $NO_2$, and 15% for HONO, in good agreement with the values found in Hendrick et al. (2014). These example vertical profiles are typical for sunny days during the measurement period where aerosol extinction and $NO_2$ were often found to peak above ground level, and HONO was found to peak at ground level. The location of the measurement site on a hill overlooking the whole city to the south, makes these profile shapes plausible, suggesting that the instrument is typically

sampling the particulate and $NO_2$ pollution plume of Melbourne. The typical HONO profile shape peaking at ground level could be a function of the greater retrieval sensitivity bias for HONO than $NO_2$ and aerosols. It could also indicate that HONO sources are ground based and highly localised, with strong HONO photolysis not allowing the same daytime vertical gradients as $NO_2$ and aerosols.

## 3   Results and discussion

### 3.1   Aerosol and trace gas retrieval tests

With no appropriate co-located measurements, the choice of a priori and forward model parameters is a source of uncertainty propagating through both the aerosol and trace gas profile retrievals (Ortega et al., 2016; Wagner et al., 2011). In this work, given the lack of independent co-located data sets, a sensitivity study was undertaken to examine the relative contribution of different a priori and forward model parameters on the final aerosol and trace gas retrieval products using the optimal estimation

method. The parameters investigated were the surface albedo, aerosol optical properties, aerosol profile shape and trace gas profile shape. The tests involved running aerosol and trace gas retrievals in HEIPRO over three sunny days of measurements at Broadmeadows, varying each parameter separately. Preliminary HEIPRO retrieval runs for March 2017 were used to determine ranges for the a priori shape parameters, including scale height ranging from 0.4-0.8 km, and ground extinction ranging





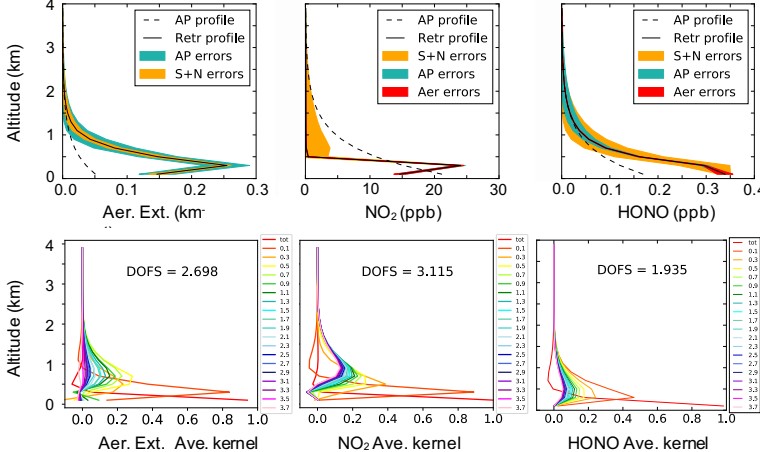

**Figure 5.** Example profile retrieval results from 1300 (local time) on 7th March 2017 at Broadmeadows, showing (left to right) the a priori and retrieved profiles, smoothing and noise error of the retrieval, averaging kernels with associated degrees of freedom for signal and retrieved diurnal aerosol optical depth or vertical column density. (a) shows aerosol extinction results, (b) $NO_2$ and (c) HONO.

from 0.04-0.08 km$^{-1}$. Ranges for the aerosol optical property tests were determined from AERONET data taken between 2003 and 2017 at six different sites around south eastern Australia, giving Angstrom exponent ranges of 0.3-1.8, asymmetry parameter 0.66-0.75 and single scattering albedo 0.7-1.0, consistent with the ranges discussed for different aerosol types over the Australian continent in Qin and Mitchell (2009). The terrain in the the MAX-DOAS instrument's field of view consists of a

grassy field in the immediate vicinity, with roadway, industrial and suburban landscapes beyond, making it difficult to estimate a uniform surface albedo value. For these tests surface albedo range limits of 0.05 and 0.2 were chosen, consistent with field observations of urban surface albedo over grass and motorways respectively (Feister and Grewe, 1995).

Figure 6(a) shows the results from different parameter tests on aerosol optical depth (AOD) and an example aerosol extinction profile from the 7th of March. Each shaded region represents the mean ± the standard deviation attributable to each test. The

vertical profile reveals a dominant contribution of a priori shape parameters (scale height and ground extinction, blue colours) to upper level uncertainty, with a 60% error contribution above 500 m. Below 500m, the influence of the shape parameters is much less significant at 10%, while the optical properties (yellow and green colours) play a more significant role with a 12% error. The observed higher sensitivity of the retrieved profiles to the a priori at high altitudes can be explained by a bias towards the a priori due to lower measurement sensitivity at these levels. The balance of these contributions suggests that a priori shape

parameters are a more significant error source than the forward model parameters, as observed in the plot of AOD over the course of 7th March, highlighting that the inherent low sensitivity of MAX-DOAS retrievals to upper levels could be greatly improved with better knowledge of the a priori shape parameters. The combined aerosol errors are included in fig. 5 alongside the retrieval smoothing and noise errors. Here it is clear that where the retrieval is most sensitive, in the lowest 500 m, the 17 % error attributable to a priori uncertainty is the most significant error source, compared to the smoothing and noise errors





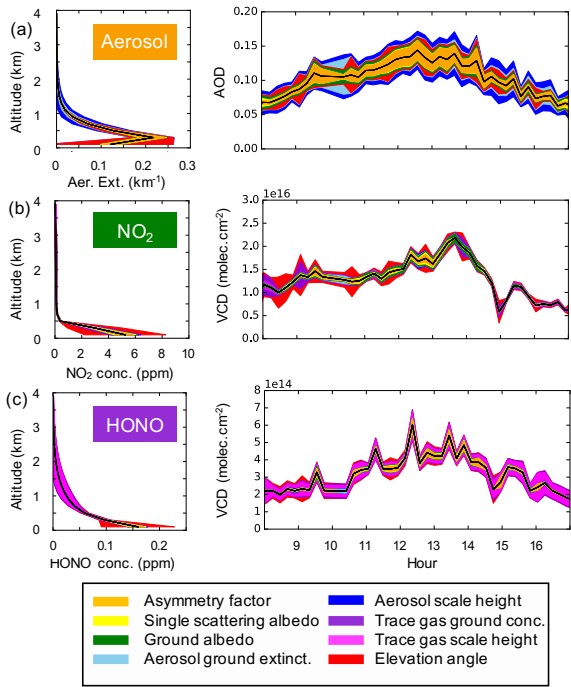

**Figure 6.** Results of a priori/forward model parameter tests for (a) an example aerosol extinction vertical profile (1 pm local time on 7th March 2017), left, and total aerosol optical depth over the course of the day on 7th March 2017, right; (b) an example vertical concentration profile of $NO_2$ and $NO_2$ vertical column density; (c) same as (b) but for HONO. Each colour represents the mean $\pm$ the standard deviation attributable to the particular test parameter as indicated by the legend.

calculated at 6 %. The influence of a $\pm 0.5°$ elevation angle uncertainty, in red, is also included. These results indicate that a $\pm 0.5°$ elevation angle uncertainty leads to a 35 % error in retrieved aerosol extinction close to the ground, and an 11 % error aloft, demonstrating the critical importance of a well calibrated elevation angle to successful MAX-DOAS retrievals.

Figures 6(b) and (c) show the results of the parameter tests on the $NO_2$ and HONO retrievals. At ground level the elevation angle term dominates the graph, highlighting the importance of elevation angle calibrations for reducing error in retrieved trace gas ground concentrations. The influence of the carry-over aerosol error and trace gas shape a priori parameters is shown to be small, between 3 and 4 % for in each case for both trace gases. This suggests that the HONO and $NO_2$ ground level concentrations are largely independent of all the aerosol and trace gas parameters, giving confidence in the success of the HEIPRO retrievals. In fig. 5 the influence of the aerosol and trace gas a priori and forward model parameters is shown to be negligible compared to the retrieval smoothing and noise errors (15 % for HONO and 10 % for $NO_2$) the lowest 500m. The high sensitivity to the ground level retrievals as demonstrated by the HONO and $NO_2$ averaging kernels, combined with the low error budget due to smoothing, noise, aerosol and shape a priori parameters in the lowest 500 m, gives confidence in the measured trace gas ground concentrations. The chemistry and atmospheric pollution implications of these trace gas concentrations are discussed in the following sections.





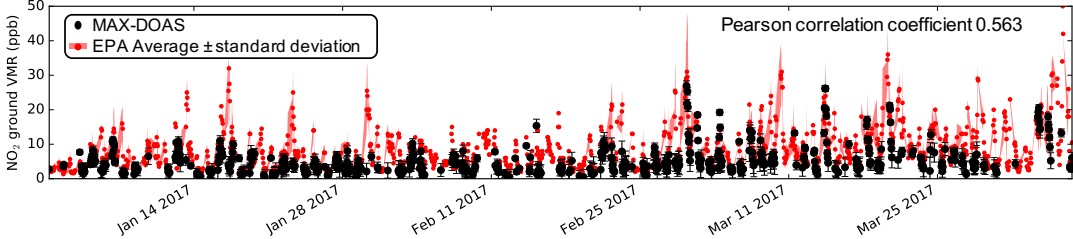

**Figure 7.** Timeseries comparison of MAX-DOAS measured $NO_2$ surface concentration at Broadmeadows (black) vs average $NO_2$ for the four Melbourne EPA monitoring sites (red). All measurements are hourly averages during daylight hours only.

## 3.2 Comparison with external data

In the case of Melbourne few options exist for the validation of surface MAX-DOAS data. For aerosol optical depth (AOD) the MAX-DOAS results were compared with AOD retrieval products from the MODIS Terra satellite, at the time of satellite overpass ($\approx$ 1400 local time daily). For a measurement period as short as three months, such comparisons are of limited

usefulness since satellite overpasses occur only once a day, and the wavelength of the MODIS 'corrected optical depth' over land was 440 nm while the MAX-DOAS AOD retrieval was at 360 nm in this study. Consistent with these limitations, while the ranges for MODIS, averaged over a 10 km spatial radius around Broadmeadows, and MAX-DOAS AOD were very similar (AOD varying between 0.05 and 0.2), the temporal correlation was weak at only 0.33. A longer sampling period and more local compatible datasets, such as PM2.5 measurements, are therefore needed for a useful validation of the MAX-DOAS aerosol

results.

While no validation data for HONO was available for Melbourne, the Victorian Environment Protection Agency (EPA) monitors $NO_2$ mixing ratios at four sites in the Melbourne metropolitan area (Footscray, Alphington, Dandenong and Altona North, as shown in fig. 1). As shown in the timeseries in fig. 7 the $NO_2$ levels measured by the MAX-DOAS at Broadmeadows are generally lower than the average of all four EPA sites over the three month measurement period, however several spikes in

the average EPA levels are captured in the MAX-DOAS results such as on January 4th and 6th, and March 1st, 2nd, 4th and 15th. Given the wide spatial range of the four EPA measurement sites, the possibility for widely varying local meteorological conditions at each site, and the fundamentally different measurement techniques, a correlation of 0.56 is a positive result for this comparison. When the local wind direction at Broadmeadows was from the south-west, correlation between the EPA stations directly to the south-west (Altona North and Footscray) the Broadmeadows MAX-DOAS $NO_2$ improved to 0.66, a positive

result which provides the strongest external validation available for these the MAX-DOAS trace gas retrievals.

## 3.3 Vertical distribution of aerosols, $NO_2$ and HONO

A three month dataset of MAX-DOAS measurements from Broadmeadows was analysed for aerosol extinction and HONO and $NO_2$ concentrations. Within this period, results were screened for the likely presence of cloud by applying a filter based on the colour index (CI). The CI was defined as the ratio between spectral intensities at 330 and 390 nm, as in Wagner et al. (2016),





for 30° elevation angle scans. The 30° scans were chosen to allow comparison of cloud filtering results with $O_4$ dSCDs, given that the sequential DOAS referencing method used returns no dSCDs for 90° elevation angles. Diurnal CI thresholds were determined for each of the three months in the campaign period by fitting a 5th order threshold polynomial to CI data from known sunny days as a function of time. Data was filtered out where the CI was less than 10 % of the threshold CI polynomial

at the given time, which was found to provide an effective filter for the short analysis period. Despite the lack of colocated external solar radiation data, cloud-flagged periods determined using the MAX-DOAS measured CI filter correlated strongly with periods of low global radiation measured at Melbourne Airport, 6 km west of Broadmeadows. Cloud filtered periods also correlated strongly with high AOD retrieval errors in HEIPRO, giving confidence in the simple empirical cloud filtering approach used here.

A timeseries of retrieved, cloud filtered surface values is shown in fig. 8 for aerosol extinction, $NO_2$ volume mixing ratio (VMR) and HONO VMR. The values shown for this example 13 day period are characteristic of the range of values observed throughout the three months, with ground level aerosol extinction 0 to 0.35 $km^{-1}$, $NO_2$ 0 to 30 ppb and HONO 0 to 0.5 ppb. The ratio HONO/$NO_2$ has been used previously to categorize emission sources of HONO with HONO/$NO_2 < 0.01$ indicating direct emission dominates HONO production, HONO/$NO_2$ 0.01 to 0.03 indicating $NO_2$ to HONO conversion at low relative

humidity and HONO/$NO_2 > 0.03$ indicating $NO_2$ to HONO conversion at high relative humidity (Wojtal et al., 2011; Hendrick et al., 2014; Qin et al., 2009). Figure 8 shows that periods of peak HONO/$NO_2$ are more commonly a function of low $NO_2$ than high HONO, and that high HONO corresponds typically to HONO/$NO_2$ around 0.03. Average diurnal cycles of HONO, $NO_2$ and HONO/$NO_2$ will be discussed in more detail in the following section.

The considerable advantage of MAX-DOAS over other methods of measuring HONO is the ability to simultaneously and

passively measure vertical distributions of HONO, $NO_2$ and aerosols. As expected given the profile retrieval sensitivity discussed above, almost all the retrieved aerosol extinction, HONO and $NO_2$ is in the lower troposphere, as shown in example vertical profile heatmaps in fig. 9. The diurnal variation of the aerosol profiles shown on these example days in early March 2017 are typical of the profiles throughout the measurement period. Aerosol extinction commonly peaked temporally in the late morning, and vertically at about 300 m, suggesting that from the elevated measurement position at Broadmeadows the MAX-

DOAS is sampling an evolving boundary layer and aerosol pollution plume over the city of Melbourne. $NO_2$ peaks often correspond well, both temporally and vertically, with aerosol extinction. HONO retrievals are most sensitive to the lowest layer and consequently retrieved HONO peaks are always at the ground level. The strong HONO vertical gradient is also consistent with previous measurements made using various techniques and platforms e.g. (Wong et al., 2012; Hendrick et al., 2014; Stutz et al., 2010; Wong et al., 2012; Young et al., 2012; Neuman et al., 2016; Li et al., 2014; Kleffmann et al., 2003; VandenBoer

et al., 2013). The presence of periods of peak aerosols and $NO_2$ above ground level could indicate that they are more likely to be influenced by longer range transport than HONO. Mixing layer heights were estimated as $H_{ML} \approx A \times (VCD/x(ppb))$, following the method of Li et al. (2013), where $x(ppb)$ is the trace gas surface mixing ratio and $A$ is the conversion factor between molecules.$cm^{-3}$ and ppb. $H_{ML}$ values for $NO_2$ and HONO correlated very strongly and were consistently shallow at around 500-700 m on sunny days. However, the correlation of vertical column density with surface mixing ratio, which is 0.89

for HONO and 0.80 for $NO_2$, suggests that surface values are a greater influence on the total column for HONO than $NO_2$.





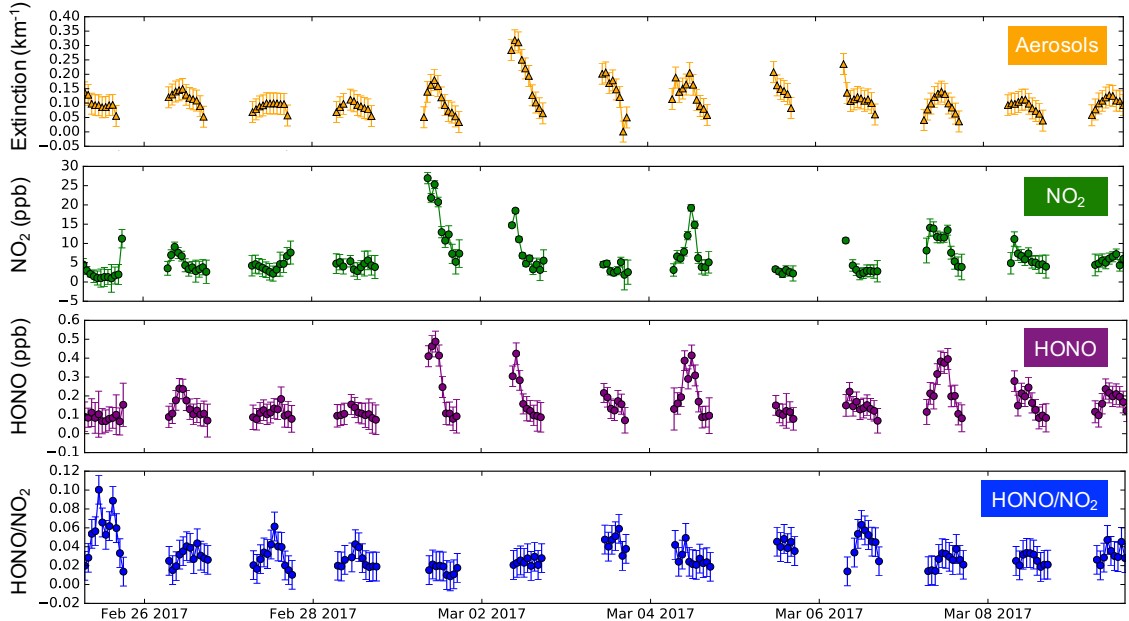

**Figure 8.** Example timeseries from 25/02/2017 to 10/03/2017 for aerosol optical depth, $NO_2$ surface concentration, HONO surface concentration and the ratio of HONO/$NO_2$ surface concentrations

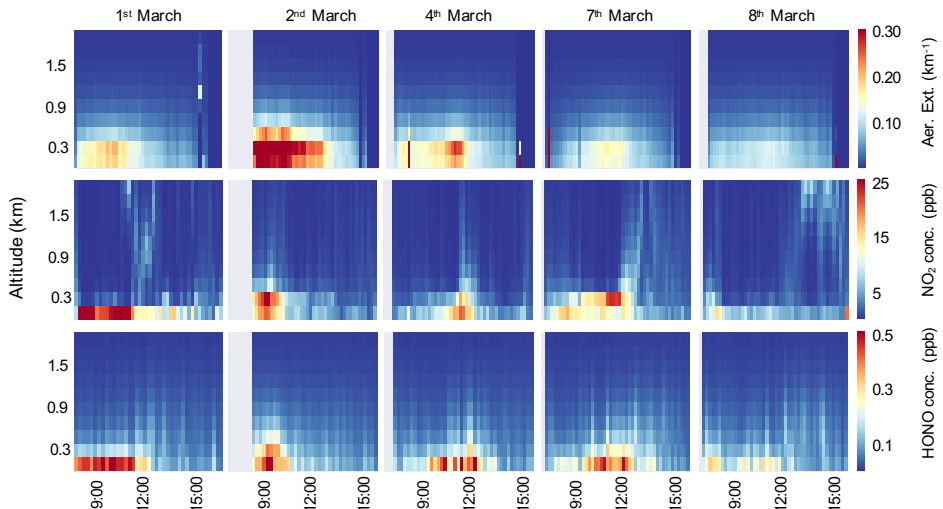

**Figure 9.** Example vertical profile timeseries, plotted as heatmaps with altitude on the y axis, for selected mostly sunny days in early March 2017. The top row shows aerosol extinction, the middle row $NO_2$ mixing ratio and the bottom row HONO mixing ratio.



### 3.4 Source distribution of aerosols, NO$_2$ and HONO

Combining MAX-DOAS measurements of aerosols and trace gases with co-located meteorological observations allows further analysis of spatial source patterns as well as vertical distributions. This is demonstrated using polar bivariate plots where the average trace gas concentrations (fig. 10(a) and (b)), and aerosol extinction (fig. 10(c)) is plotted as a function of the wind

speed and direction. While this method is not a spatial reconstruction, for this three month data set it allows estimation of the main pollution sources given that low wind speeds corresponds to localised source regions and high wind speeds to pollution transported from further afield. Aerosol extinction, HONO and NO$_2$ typically show well spatially correlated pollution peaks at low wind speed from the southerly and easterly directions. Using the local map in fig. 10 as a guide, we conclude that the major pollution sources are likely to be the Western Ring Road to the south, the busy Camp Road/Sydney Road/Ring Road

intersection to the east, and further to the east Victoria's busiest interstate highway the Hume Freeway. These results confirm that pollution is being measured by the MAX-DOAS from the expected sectors. High HONO concentration periods are more strongly influenced by both wind direction and speed than NO$_2$ suggesting that the HONO source is more strongly localised to the major road corridors.

The plots in fig. 10 are also broken down into weekdays and weekends, showing that weekend NO$_2$ decreases due to lighter

traffic are much more pronounced than for HONO. In fact during March 2017, the most polluted month of the measurement period, the NO$_2$ average daily concentration halved from 12 ppb during the week to 6 ppb on the weekend, while the HONO average daily concentration falls only from 0.20 ppb to 0.18. A similar phenomenon was found by Pusede et al. (2015) in Pasadena, suggesting that correlations and ratios of NO$_2$ to HONO needs to be treated with caution when interpreting potential HONO sources.

Surprisingly, localised aerosol sources do not decrease on weekends, and there is an overall increase in aerosol extinction from the north-east sector. This suggests aerosol extinction could be influenced by long range transport, including recycling of pollutants around the Melbourne Metropolitan area e.g. (Pearce et al., 2011), although it is unclear why this would be more apparent on weekends. It is interesting to note that while timeseries and vertical distributions of NO$_2$ and aerosol extinction are strongly correlated, the differences observed in the spatial distribution provide good evidence that there is no cross correlation

in their retrieval.

### 3.5 Periods of elevated HONO levels

During the three month measurement period, 33 days which were mostly sunny had peak HONO concentrations greater than 0.2 ppb. These periods allow analysis of the diurnal cycles of HONO, NO$_2$ and aerosol extinction, which are shown in fig. 11. The NO$_2$ diurnal cycle peaks in the early morning, consistent with morning traffic times on the nearby roadways, remains

around 8 ppb throughout the morning before decreasing until the evening traffic period around 5 pm. This diurnal cycle suggests minimal overnight accumulation of nitrogen oxides, a factor which may contribute to the very low observed HONO levels in the early morning (fig. 11(b)). In other urban studies, HONO during daylight hours typically peaks at sunrise and, despite higher than expected daytime concentrations, decreases across the course of the day. In contrast, the HONO mixing ratio measured





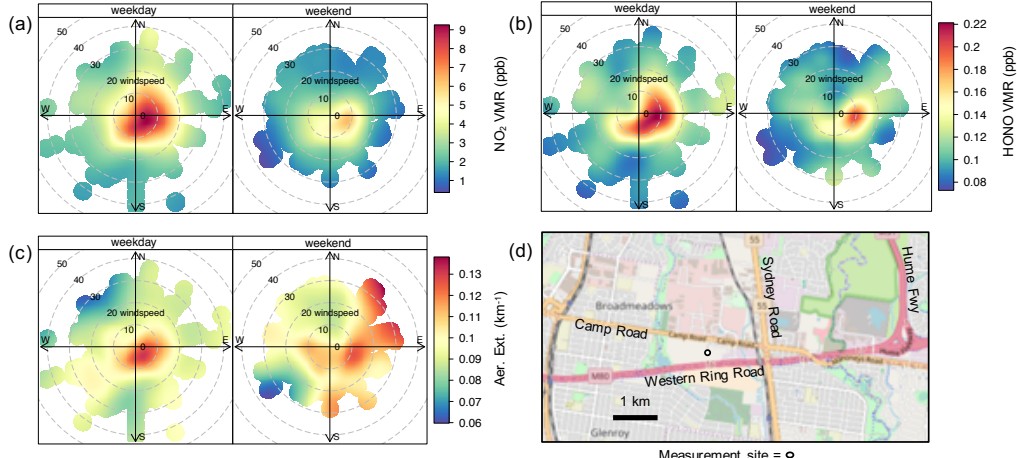

**Figure 10.** (a) Polar bivariate plot for average weekday (left) and weekend (right) $NO_2$ surface concentration as a function of wind speed and direction. (b) same as (a) but for HONO, (c) same as (a) but for aerosol extinction and (d) map indicating the location of the major road corridors near the measurement site.

here rises from an early morning minimum to a maximum averaging around 220 ppt, one hour before solar noon. Previously such daytime maxima in the HONO diurnal cycle have only been observed in rural locations for example at a rural site in Germany (Acker et al., 2006), a forested site in Michigan USA (Zhou et al., 2011), and in rural Cyprus (Meusel et al., 2016) although in each case the peak diurnal HONO value averaged significantly less than observed in Melbourne. Therefore the
5 diurnal maximum HONO measured here is unusual for an urban environment and supports the presence of a strong daytime source.

The average HONO/$NO_2$ ratio shown in fig. 11(c) exhibits a broad peak between 10 am and midday. The magnitude of the midday HONO/$NO_2$ ratio is more consistent with those measured in Cyprus (Meusel et al., 2016) and rural southern China (Li et al., 2012) than urban Beijing (Hendrick et al., 2014), bearing in mind that average peak $NO_2$ in summer in Beijing is about
10 double that measured here for Melbourne. In fact, while $NO_2$ levels are low compared to many urban centres around the world, midday HONO concentrations and HONO/$NO_2$ ratios here are comparable with midday levels reported in Beijing (Hendrick et al., 2014), London (Lee et al., 2016), Pasadena (Pusede et al., 2015) and Houston (Wong et al., 2012).

The implication of the high daytime HONO levels for the tropospheric oxidation capacity was assessed by comparing the OH radical production rates from HONO photolysis (R3) and ozone photolysis (R1, R2). These are given, respectively, by:

$$P(\text{OH})_{\text{O}_3} = 2 \times f \times J(\text{O}(^1\text{D})) \times [\text{O}_3] \tag{3}$$

$$P(\text{OH})_{\text{HONO}} = J(\text{HONO}) \times [\text{HONO}] \tag{4}$$





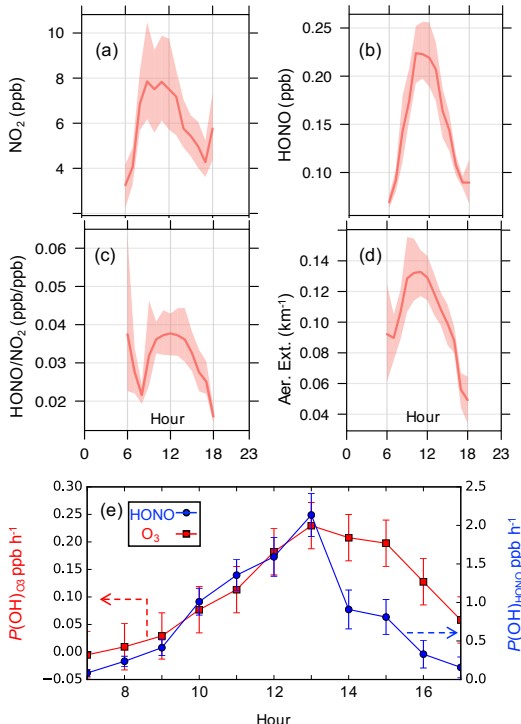

**Figure 11.** Data from 33 days in the measurement period with high daytime peak HONO. (a) Diurnal cycle plot for 1 hourly averages of HONO surface concentration at Broadmeadows. Solid line shows average value, the shaded region represents the 95 % confidence interval. (b) Same as (a) but for the $NO_2$ surface concentration. (c) Same as (a) but for the surface concentration ratio HONO/$NO_2$. (d) Same as (a) but for the aerosol ground extinction. (e) Diurnal cycle of OH radical production *P(OH)* from ozone (red, primary y-axis) and HONO (blue, secondary y-axis) photolysis, calculated for an example day with 30 ppb ozone, and the HONO diurnal cycle from 7th March 2018 which peaked near midday at around 400 ppt.

In equation 3, $f$ is the fraction of $O^1D$ reacting with water vapour to form OH (Finlayson-Pitts and Pitts Jr, 1999). $[O_3]$ is the ozone concentration in ppb which was taken from averaged EPA measurements of surface ozone concentration around Melbourne, [HONO] was the surface HONO concentration in ppb calculated from the MAX-DOAS retrieval. The photolysis rates $J(O(^1D))$ and $J(HONO)$ were calculated using the TUV radiation model (Madronich and Flocke, 1999). Photolysis

5    rates were simulated for 7th March 2017, a clear sunny day with a HONO midday peak of 400 ppt (fig. 8), and Melbourne EPA average $O_3$ concentration 27 ppb in the late afternoon. The aerosol optical depth was fixed at 0.15, and the total ozone column was fixed at 270 D.U. consistent with zenith DOAS measurements using the same MAX-DOAS instrument. The TUV simulated $J(O(^1D))$ values were consistent with the empirical parameterisation of Wilson (2015) for $J(O(^1D))$ at Cape Grim in north-western Tasmania. The resulting $P(OH)$ diurnal values are presented in fig. 11(e), showing that the peak OH

10   production rate in the middle of the day is around 0.2 ppb h$^{-1}$ from $O_3$ photolysis, and around 2 ppb h$^{-1}$ from HONO





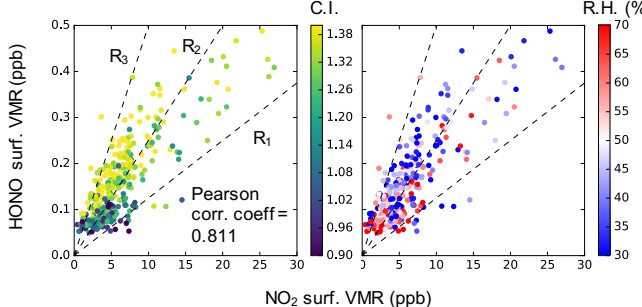

**Figure 12.** Scatter plots of (a) HONO ground concentration vs $NO_2$ ground concentration coloured by colour index (CI), showing dependence of high HONO values on solar radiation. Note that CI is the ratio of intensities measured by the MAX-DOAS instrument at 330 and 390 nm. The cloud filtered CI correlated strongly with global irradiance measured 6 km to the west at Melbourne Airport. (b) HONO ground concentration vs $NO_2$ ground concentration coloured by relative humidity. Dashed lines represent the ratios $R_1$ HONO/$NO_2$ = 0.0125, $R_2$ HONO/$NO_2$ = 0.025 and $R_3$ HONO/$NO_2$ = 0.05.

photolysis. This suggests that that HONO levels at 400 ppt in the middle of the day can increase the OH radical production by an order of magnitude, significantly increasing the local tropospheric oxidative capacity in Melbourne. For the 33 days with high daytime HONO peaks, the average diurnal cycle peak of [HONO] at 220 ppt corresponds to a source of OH radicals around 1 ppb h$^{-1}$, five times greater than from ozone photolysis.

## 3.6 Possible daytime HONO sources

The HONO diurnal profile observed in this campaign matches temporally with the diurnal profile of the missing HONO production source calculated in both rural e.g. (Meusel et al., 2016) and urban e.g. (Wong et al., 2012; Pusede et al., 2015). Unfortunately, no hydroxyl or nitric oxide (NO) measurements were available during the campaign precluding the calculation of expected photo-stationary state HONO concentrations and the magnitude of relative source contributions. Nevertheless comparing the MAX-DOAS measurements with co-located meteorological data provides insight into possible source mechanisms for the high daytime HONO concentrations. The heterogeneous conversion of $NO_2$ on wet surfaces according to reaction R5 has been suggested as a primary HONO source pathway (Wong et al., 2012) especially during the night when there are no OH radicals available to form HONO via reaction R4. Values of the HONO/$NO_2$ ratio between 0.01 and 0.03 have been found when the conversion via reaction R5 proceeds in low relative humidity environments while HONO/$NO_2$ > 0.03 indicates conversion at high relative humidity. The average midday HONO/$NO_2$ ratio averages around 0.035 (fig. 11(c)) in this case, however fig. 12(b) shows that most of the data points where HONO/$NO_2$ is between 0.025 and 0.05 correspond to low relative humidity. The overall correlation between HONO surface concentrations and relative humidity is weak and negative at -0.31 further indicating that reaction R5 cannot explain the high daytime HONO.

Even if reaction R5 cannot explain the observed HONO levels, the strong correlation of 0.81 between HONO and $NO_2$ surface concentrations (fig. 12) suggests that $NO_2$ is implicated in some other way. One mechanism could involve photo-





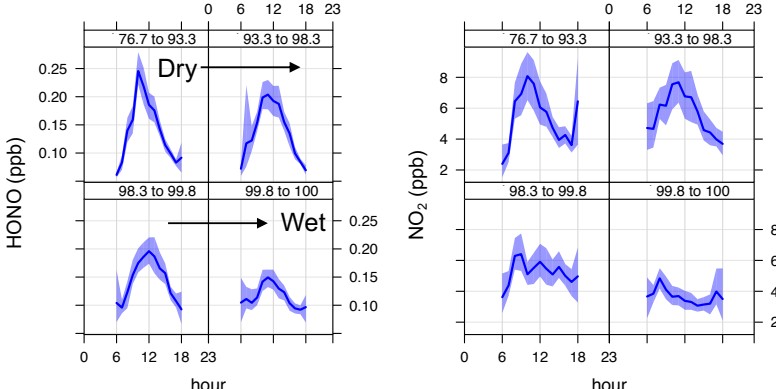

**Figure 13.** (Left) HONO and (right) $NO_2$ diurnal cycles over the three month measurement period, divided into four rainfall index bins where lowest rainfall index indicates days likely to have lowest soil moisture, and highest rainfall index indicates days with highest soil moisture.

enhanced uptake and conversion of $NO_2$ on aerosols, such as organics, as proposed by George et al. (2005). Only moderate temporal correlations of 0.58 and 0.49 between aerosols and HONO were found for surface and total column amounts respectively. In addition, HONO and aerosol vertical distributions were often different as indicated in figures 5 and 9, indicating that aerosol-mediated $NO_2$ conversion cannot explain the observed high daytime HONO levels. This argument also suggests that

photodissociation of particulate nitrate, which was found to be a HONO source with a diurnal peak before midday by Ye et al. (2017), is not the major daytime HONO source here.

    While the diurnal cycles of HONO and HONO/$NO_2$ indicate a more 'rural' type of HONO production, the high HONO to $NO_2$ correlation in Melbourne matches what is expected in an urban setting (Hendrick et al., 2014). The correlation holds as a function of wind speed, with both HONO and $NO_2$ being localised (fig. 10) although HONO is more dependent on wind

direction. As discussed in the source distribution section above, the correlation does not hold between weekends and weekdays and these combined factors suggest that while plausible photo-activated, ground based $NO_2$ conversion mechanisms exist, the correlation does not necessarily entail high $NO_2$ to HONO conversion.

    Therefore other mechanisms, disconnected from $NO_2$, may be contributing to the daytime HONO formation. Processes at the ground surface have been suggested to provide strong daytime HONO sources, including nighttime deposition of HONO

or gaseous atmospheric acids to a surface reservoir followed by daytime re-emission (VandenBoer et al., 2014, 2015), and photoactivated reactions involving humic acids (Stemmler et al., 2006) and soil nitrites (Su et al., 2011). Such a ground-based sources match both the strong vertical HONO gradients and diurnal profile of the daytime HONO observed in Melbourne, suggesting that these mechanisms could explain the presence of HONO in the middle of the day. Lee et al. (2016) concluded that in London, the highly urbanised environment surrounding the measurement site meant soil-based HONO contributions were un-

likely to contribute to the observed high daytime HONO budget. In contrast, the outer-suburban Broadmeadows measurement





site is surrounded by a variety of surfaces including vacant fields, wide grassed road verges and parkland and consequently soil-based processes must be considered in the possible daytime HONO formation mechanisms. Meusel et al. (2018) showed that in Cyprus HONO emissions from soil biocrusts could explain the daytime HONO budget. They also found that NO was emitted from soil biocrusts and that maximum HONO and NO emission was observed for bare soil with moisture contents

around 25 %. With no soil measurements available during this campaign, a simple approximation for soil moisture content was made by considering how HONO and $NO_2$ diurnal cycles changed in relation to the timing and amount of rainfall. An empirical rainfall index was defined for each day according to $RI = 100 - D_{pr}/R_{pd}$, where the rainfall index on a day with any rain is arbitrarily defined as $RI = 100$, $D_{pr}$ is the number of days since the previous rainfall event (i.e. $D_{pr} = 0$ if raining today, $D_{pr} = 1$ if it rained yesterday, and so on) and $R_{pd}$ is the amount of rainfall, in mm, in the most recent rainfall event.

Rainfall on consecutive days was defined as one rainfall event, with $R_{pd}$ being the sum of all rainfall in that event. $RI$, and hence assumed soil moisture, was therefore a maximum, 100, on rainy days decreasing to a minimum when (a) it had not rained for many days and or (b) the previous rainfall amount was small. Figure 13 shows that the average HONO diurnal cycle daytime maximum was much higher when the soil was likely to be dry, rather than wet, suggesting that daytime HONO in Melbourne may indeed depend on soil moisture. These findings are consistent with the low HONO emissions from biocrusts

at very high soil moisture content as discussed in Meusel et al. (2018). In that paper, very low soil moisture were predicted to result in decreased HONO production efficiency. Such a decrease in peak HONO values is not observed here, possibly due to the frequency of rainfall events during the campaign or the nature of the local soil biocrusts. Interestingly the same trend was observed for $NO_2$ which suggests that periods with $NO_2$ peaking in the middle of the day may be attributable to soil-based NO production rather than traffic emissions. This is highlighted by the $NO_2$ diurnal cycles in the two wettest $RI$ bins which show

early morning increases consistent with morning peak traffic times, but do not go on to match the high midday levels seen in the two driest $RI$ bins. These preliminary findings point to an important role for soil-based emissions in the observed level of nitrogen oxides, however NO, soil moisture, chemistry and biology measurements, as well as a longer measurement period, will be necessary to determine the responsible soil based mechanisms in Melbourne.

## 4    Conclusions

This study reports on the first MAX-DOAS measurements from Melbourne, Australia. A detailed uncertainty analysis of the retrieval a priori and aerosol parameters, combined with HONO DOAS fitting window optimisation, provide confidence in retrievals of aerosol extinction, $NO_2$ and HONO using the HEIPRO algorithm. The $NO_2$ results are comparable to the EPA air quality monitoring carried out around Melbourne, with average maximum $NO_2$ levels for the three month measurement period around 8 ppb. Despite the moderate to low $NO_2$ pollution levels, high daytime HONO was commonly recorded with

peak values in the late morning around 220 ppt. Such a consistent daytime HONO diurnal peak has previously been reported only in rural areas, and matches previously calculated diurnal profiles for a HONO missing source. While strong vertical and temporal correlations between HONO and $NO_2$ exist generally, the correlation does not hold between weekends and weekdays, suggesting that some of the high HONO levels are decoupled from $NO_2$ conversion. Strong relationships between





solar radiation and HONO and strong HONO vertical gradients support previous theories that the missing HONO source is photolytically active and ground based. Furthermore, a dependence of both $NO_2$ and HONO on the time since significant rainfall suggests that soil moisture may be playing an important role in the local nitrogen oxide chemistry. These findings suggest that HONO may be significantly increasing the local tropospheric oxidation capacity, providing an OH radical source 5-10 times greater in the middle of the day than from ozone photolysis alone. Future studies should further explore the relationship of different species in the oxidative cycle to the HONO and OH predicted from this work. Measurement of absolute OH radical concentrations and other nitrogen oxide species such as nitrates, would allow a more detailed analysis of the expected HONO pathways and mechanisms for the strong daytime HONO source observed.

*Code availability.* TEXT

*Data availability.* TEXT

*Code and data availability.* TEXT

*Author contributions.* TEXT

SR and MT maintained the MAX-DOAS instrument collecting the data. RR conducted the data analysis and drafted the manuscript. RR, SW, NJ, UF and RS contributed to developing the scientific direction, analysis protocols and writing the manuscript.

*Competing interests.* TEXT

The authors declare that no competing interests exist.

*Disclaimer.* TEXT

*Acknowledgements.* RR wishes to acknowledge helpful discussions with Prof. Peter Rayner on atmospheric inverse methods and Dr Johannes Lampel on DOAS fitting methods, as well as Paul Torre and the Victorian Environment Protection Agency for making data air quality available. RS and RR acknowledge support from Australian Research Council's Centre of Excellence for Climate System Science





(CE110001028) and Australian Research Council's Discovery project: Tackling Atmospheric Chemistry Grand Challenges in the Southern Hemisphere (DP160101598).



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
