# Peer review of "Daytime HONO, NO2 and aerosol distributions from MAX-DOAS observations in Melbourne"

_Atmospheric Chemistry and Physics, 2018_

## Referee Comment (RC2) · Anonymous Referee #2 · 19 Jun 2018

This work comprehensively presents an approach for MAX-DOAS retrievals for HONO, NO2, and aerosol vertical distributions in the absence of collocated measurements. The manuscript describes the approach taken to calculate the vertical profiles, clearly presents the underlying assumptions with literature reference for context and tests the potential error in those assumptions through a sensitivity analysis. This analysis allows the Authors to present vertical profiles of the two trace gases and aerosol over the urban area of Melbourne, Australia. Unlike most urban environments, daytime peaks in HONO concentrations were observed that typically have been reported for rural environments. The authors present suggestions as to the potential sources of this observation within the context of several recent findings in the literature. Overall, this

is a solid manuscript, which is quite nearly enough work for two separate publications and will be suitable - with minor revisions suggested below - for publication in Atmospheric Chemistry and Physics. Some lengthy areas and a number of figures could be clarified/removed from the manuscript into a Supporting Information document to increase the focus of the work. Some literature regarding daytime HONO release from soils is absent from the discussion.

There is a single major revision to this work that should be made regarding regressions. It is not clear what regression approach the Authors used, but atmospheric datasets typically require accounting of error in both measures (e.g. (Wu and Yu, 2018)). The authors present a Pearson correlation coefficient in Figure 7 and discuss correlations heavily in their discussion from there on. This suggests that a linear least-squares analysis was used, which assumes error in the ordinate alone. The authors should clarify this and present appropriate regression metrics (e.g. slope and regression coefficient) when discussing correlations. Much of the discussion surrounding these comparisons is qualitative while the wording suggest that quantitative evaluations have been made. Including this quantitative information will strengthen the discussion.

Minor comments throughout the manuscript are as follows:

Reference to figures throughout the manuscript should be capitalized as 'Fig. X' instead of 'fig. x'

Page 3, Lines 30-35: Interferences from clouds, as presented in the discussion, should be included in the drawbacks here.

Page 4, Line 30: repetition of 'and' to be corrected

Page 5, Line 11: differential slant column density is presented here and presumably is the source of the 'dSCD' term used later in the paper. Please define here, if this is correct.

Page 6, Figure 2: Keep the traces in panel c) consistent with the caption. Suggest

switching O3 and HCHO in the caption to be in the same order as the traces.

Page 6: Figure 2 and Table 1 could be moved to a supporting information document to reduce manuscript length.

Page 7: Figure 3 could be moved to a supporting information document to reduce manuscript length. Figure 4 could be easier to interpret if the time axis is consistent with the others in the manuscript. Two hour time intervals here, and in other diurnal plots, would provide the most detail without becoming cluttered. Figure 4 could benefit from being presented with larger panels if the other figures and table are moved to supporting documentation.

Page 8: Section 2.3 is quite long and is detailed for the manuscript, but feels like a lot of detail has also been left out. A suggestion here would be to simplify and condense this section further to improve its clarity (e.g. Equation 1 is not accessible to those not familiar with all of the literature in this section and could be considered superfluous along with many of the details) or some of the details could be moved to supporting documentation and expanded upon in the interests of allowing greater ease of reproduction of this detailed work.

Page 8, Line 14: K presented as a weighting function matrix does not appear in either Equation 1 or Equation 2. Please revise.

Page 9, Lines 9-23: These are results. Suggest relocating to the results and discussion section.

Page 10, Figure 5: There are acronyms (or short-hand notation) being used in the upper row of panels which are not defined in the caption. Please do so. In the bottom row of panels, there are 20 different lines presented in each panel and the values for each are very small in the legend. Are all of these necessary or can half of them be removed without undermining the findings? It would allow all panels in this figure to be increased in size and make it easier to read. Finally, panels (a) and (b) are not labeled

here. Please add these.

Page 10, Line 4: repetition with 'the'

Page 10, Lines 13-14: 'by a bias towards the a priori due to lower measurement sensitivity at these levels' This explanation is unclear. Please consider revising to improve the clarity here.

Page 11, Figure 6: This figure could be moved to a supporting document. The panels are alphabetically labeled, so the text boxes for each can be removed. VCD is not defined in the caption. Also add a note regarding the exponent terms for NO2 and HONO VCD values as they may be easily missed.

Page 11, Line 10: 'the lowest 500m' should be 'in the lowest 500 m'

Page 11, Lines 11-14: Is the 'high sensitivity' instrumental sensitivity or retrieval sensitivity? Use of the word sensitivity here is a bit unclear and the retrieval sensitivity might be more intuitively termed 'potential error' or 'estimation error' or simply 'error'.

What is a 'low error budget'? This wording is not consistent with the rest of this section. Please clarify.

Page 12, Figure 7: From here on forward, the regression analyses should be clearly presented. A useful quantity that would have been obtained in the analysis here is the slope, which gives some indication of the bias that is discussed qualitatively. Such bias is expected given that the in-situ monitors are located at ground level (on top of sources), while the MAX-DOAS is observing more dilution of those sources. In the caption of this figure there is a reference to 'surface concentration' measured by the MAX-DOAS, but isn't the lowest elevation angle somewhere between 100 - 500 m above ground level? This isn't terribly clear and could help bridge these observations with ground observations more easily.

Page 12, Line 9: 2.5 should be a subscript. In addition, wouldn't ceilometer or LIDAR measurements be more useful in validating the MAX-DOAS aerosol results?

Page 12, Line 17: The technique used by the EPA has not been presented and should be added as supporting instrumentation details in section 2.1. Presumably these are chemiluminescent analyzers with molybdenum converters?

Page 12, Line 19: What is the slope of the comparison? What type of regression was used?

Page 13, Lines 6-9: Add appropriate quantitative regression data here, along with coefficient values that justify selection of wording such as 'correlated strongly'. The direction of the correlation is also important. Was the relationship a positive or negative one in each of these cases? Please provide these quantitative details.

Page 13, Lines 14-17: These ratios have been derived from measurements made at night or from tunnel studies and their applicability to the interpretation of daytime data is questionable. During the day, the longer lifetime of NO2 relative to HONO could result in the observed diurnal pattern of HONO/NO2 by simple boundary layer mixing processes diluting the surface NO2 while the surface HONO source does not change (i.e. it could be independent of NO2, as suggested by the weekend dataset). It may be worthwhile to discuss this further and carry it into the later discussion or to remove HONO/NO2 as a suitable daytime metric entirely.

Page 13, Lines 33-35: Quantitative values for 'correlated strongly'. This is consistent with literature reports of surface processes dominating over aerosol NO2 conversion. Please cite some examples of this.

Page 14, Figure 8: This figure could be moved to the supporting information document. Remove text boxes on each panel. Label each alphabetically. If necessary, clarify what is on each panel in the caption. Reduce the number of labeled ticks on each ordinate axis.

Page 14, Figure 9: Change time axis to two-hour intervals. The date format on top of each column is different from that in Figure 8. Keep date formats consistent throughout

the manuscript and consistent with ACP guidelines. Remove 'conc.' from the HONO and NO2 labels. They are correctly identified as mixing ratios in the caption.

Page 15, Line 7: Delete 'well'

Page 15, Lines 27-28: These daytime values are higher than might be expected given that the measurement is being made through a large volume and from the NO2 intercomparison. This would suggest in-situ HONO measurements might exceed 0.5 ppb. How does the > 0.2 ppb HONO value compare to other reports in urban and rural environments?

Page 15, Lines 32-33: The authors discuss that HONO does not peak in the early morning during their daylight observations. Are there examples of MAX-DOAS observations capable of seeing the previous night's HONO prior to photolysis? What are the vertical resolution differences between these MAX-DOAS measurements and how might that impact the observations (i.e. if the 'surface' bin is deeper than other observations, you'd expect to observe lower levels). Also, are there limitations in the MAX-DOAS measurement near sunrise with the instrumental orientation that could result increase the error in capturing a quantitative absorption signal for HONO at this time?

Page 16, Figure 10: Consider moving this figure to a supporting information document. The discussion does a good job of conveying the information presented here.

Page 16, Lines 1-5: It would improve the discussion to report the daytime maximum mixing ratios observed in these other locations, for context.

Page 17, Figure 11 (and other similar instances): The caption does not describe the panels correctly here. Further, the caption description for the similar panels can be improved by changing the phrase directed for the first panel to the following: 'Diurnal cycle plots for the 1 hourly averages of (a) NO2, (b) HONO, (c) HONO/NO2, and (d) aerosol extinction surface values at Broadmeadows'. The alphabetical indicators for

each panel should be displayed outside the axes throughout this figure.

Page 18, Line 7: The citations here are not in the proper format to present via 'e.g.'. This is also a hanging sentence. Please correct it.

Page 18, Lines 11-18: These ratio values continue to be potentially misleading. Suggest careful revision or even removing this part of the discussion since intensive chemical description of the HONO/NO2 ratio under sunlit conditions has not been well-established.

Page 18, Line 20: 'suggests that NO2 is implicated in some other way'. The weekend data presented here suggests that this may not be true OR that the mechanism is NO2-saturated. See some discussion of this in (Pusede et al., 2015).

Page 19, Figure 13: The data presented here suggest that there is a suppression of HONO daytime surface flux due to increased soil water content. There are a few instances of this hypothesis being tested under laboratory and field conditions that may be worth mentioning here (Donaldson et al., 2013;Donaldson et al., 2014;Oswald et al., 2013;Scharko et al., 2015;Su et al., 2011;Weber et al., 2015). Comparison to the microbial pathways, reversible partitioning, and surface adsorption/dissolution could all enhance the discussion.

Page 19, Lines 1-2: The literature has been clear on the aerosol surface area conversion of NO2 to HONO being a minor daytime production route for some time. Suggest including some references to the literature that have demonstrated the phenomenon here in support of your findings.

Page 20, Line 17: This instance of biocrust discussion should be expanded if there are local biocrusts near the observation site, and generally throughout the Melbourne area. The established literature on this, coupled to anything known about regional biocrust microbial composition, may facilitate a stronger capacity to speak on this potential daytime HONO source instead of speculating.

References

Donaldson, M. A., Berke, A. E., and Raff, J. D.: Uptake of gas phase nitrous acid onto boundary layer soil surfaces, Environmental Science & Technology, 48, 375-383, 10.1021/es404156a, 2013.

Donaldson, M. A., Bish, D. L., and Raff, J. D.: Soil surface acidity plays a determining role in the atmospheric-terrestrial exchange of nitrous acid, Proceedings of the National Academy of Sciences, 111, 19472-18477, 10.1073/pnas.1418545112, 2014.

Oswald, R., Behrendt, T., Ermel, M., Wu, D., Su, H., Cheng, Y., Breuninger, C., Moravek, A., Mougin, E., Delon, C., Loubet, B., Pommerening-Röser, A., Sörgel, M., Pöschl, U., Hoffmann, T., Andreae, M. O., Meixner, F. X., and Trebs, I.: HONO emissions from soil bacteria as a major source of atmospheric reactive nitrogen, Science, 341, 1233-1235, 10.1126/science.1242266, 2013.

Pusede, S. E., VandenBoer, T. C., Murphy, J. G., Markovic, M. Z., Young, C. J., Veres, P. R., Roberts, J. M., Washenfelder, R. A., Brown, S. S., Ren, X., Tsai, C., Stutz, J., Brune, W. H., Browne, E. C., Wooldridge, P. J., Graham, A. R., Weber, R., Goldstein, A. H., Dusanter, S., Griffith, S. M., Stevens, P. S., Lefer, B. L., and Cohen, R. C.: An atmospheric constraint on the NO2 dependence of daytime near-surface nitrous acid (HONO), Environmental Science & Technology, 49, 12774-12781, 10.1021/acs.est.5b02511, 2015.

Scharko, N. K., SchuÌĹtte, U. M. E., Berke, A. E., Banina, L., Peel, H. R., Donaldson, M. A., Hemmerich, C., White, J. R., and Raff, J. D.: Combined flux chamber and genomics approach links nitrous acid emissions to ammonia oxidizing bacteria and archaea in urban and agricultural soil, Environmental Science & Technology, 49, 13825-13834, 10.1021/acs.est.5b00838, 2015.

Su, H., Cheng, Y., Oswald, R., Behrendt, T., Trebs, I., Meixner, F. X., Andreae, M. O., Cheng, P., Zhang, Y., and Pöschl, U.: Soil nitrite as a source of atmospheric HONO and OH radicals, Science, 333, 1616-1618, 10.1126/science.1207687, 2011.

Weber, B., Wu, D., Tamm, A., Ruckteschler, N., Rodriguez-Caballero, E., Steinkamp, J., Meusel, H., Elbert, W., Behrendt, T., Sörgel, M., Cheng, Y., Crutzen, P. J., Su, H., and Pöschl, U.: Biological soil crusts accelerate the nitrogen cycle through large NO and HONO emissions in drylands, Proceedings of the National Academy of Sciences, 112, 15384-15389, 10.1073/pnas.1515818112, 2015.

Wu, C., and Yu, J. Z.: Evaluation of linear regresion techniques for atmospheric applications: the importance of appropriate weighting, Atmospheric Measurement Techniques, 11, 1233-1250, 10.5194/amt-11-1233-2018, 2018.

---

## Author Comment (AC1) · 9 Aug 2018

AUTHOR RESPONSES IN BLUE ITALIC TEXT

1. This paper reports measurements of HONO, NO$_2$ and aerosol using a MAX-DOAS instrument in the city of Melbourne, Australia. It shows enhanced levels of HONO, often peaking in the middle of the day, which would not typically be expected. The authors postulate a ground based photoactivated source of HONO, using evidence based on the dependence of high HONO levels since rainfall, combined with the observed diurnal profiles. It is an interesting paper with potentially significant results in terms of the effect of HONO as an OH source and hence on atmospheric oxidizing capacity. It

is well written with good, easy to see figures. However, it suffers from the fact that no concurrent other measurements were made, making a full analysis of the effect of HONO on the chemical processes happening difficult. Hence many of the conclusions drawn are based on a bit of speculation which is not ideal. However, the data is of such interest (especially as it contains vertical profiles of HONO) that I do believe it should be published subject to some extra analysis. I realise there is no way to go back and back the extra measurements required but I think there are things that could be done to improve the analysis and conclusions.

*We thank the reviewer for their suggestions that lead to improvements of the reactive nitrogen chemistry and exploitation of vertical information provided by the MAX-DOAS, significantly improving the manuscript.*

2. One of the great advantages of the MAX-DOAS measurement is that it gives a vertical profile of HONO. Often, measurements are only made at the ground and as HONO is so short lived and postulated sources are often surface based, it is possible that the effect of HONO as an OH source in the entire boundary layer is overestimated. Here, the authors calculate P(OH) from HONO and ozone photolysis and show that in the daytime, OH from HONO is an order of magnitude more than from ozone. However could they do this for the entire vertical profile measurements and hence provide a comparison between the two sources of OH for the entire boundary layer? This would provide an interesting contrast to just looking at the surface data.

*In order to compare the OH radical production throughout the troposphere, we have accessed ozone-sonde data collected at the Broadmeadows site which provides temperature, relative humidity and ozone mixing ratio profiles throughout the troposphere (and stratosphere). Using the TUV model, photolysis rates J(HONO) and $J(O^1D)$ were calculated as a function of height through the troposphere. Hence equations 3 and 4 in the manuscript could be used to calculate the OH production rates P(OH) in ppb $h^{-1}$ throughout the troposphere, which are shown in a new figure (now Fig. 9).*

*It should be noted that in the course of these calculations a mistake was found in the original surface P(OH) calculation and therefore (previously) Fig. 11(e) has been updated. The wrong ozone mixing ratios were used in calculating $P(OH, O^1D)$ meaning that this was underestimated. Accordingly, using the HONO diurnal cycle from 7th March and the appropriate ozone mixing ratios taken from the average EPA values (peaking at 27 ppb) around the city of Melbourne, the OH production due to HONO exceeded that due to ozone by a factor of 4 rather than a factor of 10. The highlighting of the OH production source due to HONO in the abstract of the manuscript has also been changed from reporting a source "up to ten times stronger" to "up to four times stronger".*

*HONO has previously been observed to be the dominant primary OH production mechanism in urban areas (e.g. Ren et al. (2003); Elshorbany et al. (2009)) using in situ measurements and modelling of surface mixing ratios. Given that the MAX-DOAS technique provides vertical profiles of HONO, the calculation of vertical OH production profiles due to HONO photolysis is possible. With co-located ozone sonde measurements at the Broadmeadows site, primary OH production has been compared across the lowest 8 km of the troposphere in Fig. 9(b). Ozone sonde data has been averaged across all measurements (17 midday measurements, approximately weekly) during the MAX-DOAS measurement period (21 December 2016 to 7th April 2017). This is compared with the average midday HONO profile from the 33 days with peak HONO greater than 0.2 ppb in the lowest retrieval layer. It was assumed that there were no HONO sources above the MAX-DOAS top retrieval height (4 km). HONO Fig. 9(b) shows that while OH production is dominated close to the ground by HONO photolysis, ozone photolysis is dominant above 1 km and will therefore be the dominant OH radical source throughout the whole troposphere. This demonstrates that considering only surface values can give a distorted picture of the relative importance of different radical sources, and highlights the ability of the MAX-DOAS technique to provide important vertically resolved information on tropospheric oxidation chemistry."*

3. The authors should also make some comment about other radical sources and how these may compare to the primary OH production from HONO and ozone photolysis (even if they have to estimate what concentration of other species may be).

*We have added brief comments on the OH chemistry in response to this comment and point 5 (below). However, in this work we have restricted our study focus to the reactive nitrogen cycle and hence feel that discussion of other radical sources (i.e. VOCs) falls outside the scope of the present paper. Addressing this question with sufficient detail would require emissions data and/or in situ measurements for VOCs, among other species, which are currently lacking for Melbourne. However, given the ability of the MAX-DOAS to measure both ozone and formaldehyde, a key VOC oxidation product, we aim to address this question in future work with an extended MAX-DOAS dataset.*

4. I think showing correlation between HONO and $NO_2$ at different altitudes as well as just at the ground (as in figure 12), would provide some information as to a potential HONO source. Presumably the correlation should get less with increasing height if the HONO source is some form of ground based $NO_2$ conversion.

*This is an interesting idea which was partially explored in the manuscript (previously at Page 13, lines 30-35) when considering the relationship between the retrieved total column and surface values for HONO and $NO_2$. Here it was found (now using improved regression analysis following comments from Reviewer 2) that the correlation between surface mixing ratio and total column was stronger for HONO than for $NO_2$ potentially indicating that the HONO source was more likely dominated by the surface than $NO_2$.*

*To extend this to the discussion of potential HONO sources as the Reviewer suggests, the correlation between $NO_2$ and HONO mixing ratios in different retrieval layers was calculated and does indeed decrease with increasing altitude (the table appears in the Supplementary Information Document). However, the interpretation of this finding is not straightforward. While this could indicate conversion of $NO_2$ at the ground level is contributing to the observed HONO, the same result could equally be interpreted as*

*being due to the strong vertical HONO gradient due to its shorter lifetime. Furthermore, comparison of correlations at different altitudes from the MAX-DOAS retrieval is complicated by the fact that except for the total column and the lowest retrieval layer, the sensitivity of the retrieval to the measurements (see averaging kernels) is different for $NO_2$ and HONO. Furthermore, the work conducted to address point 6 below suggest that $NO_2$ ground conversion cannot bridge the missing daytime HONO source gap, and this point has been added to the discussion.*

*In the text of the manuscript, the question has been addressed at (previously) Page 18, lines 19-20, which previously read: "Even if reaction R5 cannot explain the observed HONO levels, the strong correlation of 0.81 between HONO and $NO_2$ surface concentrations (fig. 12) suggests that $NO_2$ is implicated in some other way."*

*Updated text: "The strong correlation coefficient of 0.81 between HONO and $NO_2$ mixing ratios in the lowest retrieval layer (Fig. 10) suggests that $NO_2$ may implicated in the daytime HONO production. The correlation coefficient decreases with increasing altitude (see Table S3 in the Supplementary Information) which could indicate that conversion of $NO_2$ at the ground level is contributing to the observed HONO. However, caution should be taken in interpreting this result since the shorter lifetime and hence expected stronger vertical gradient of HONO compared to $NO_2$ would also lead to a decreasing correlation with altitude. Furthermore, given that the PSS calculation includes the strong $NO_2$ ground conversion rate in Lee et al. (2016) and still cannot replicate the average HONO diurnal profile, photolytic ground $NO_2$ conversion cannot be the dominant daytime HONO producer in Melbourne."*

5. It is a shame there are no NO measurements to allow a steady state and thus a 'missing' HONO concentration to be calculated. However, the authors could make some broad estimate of NO based on their $NO_2$ measurement and at least a rough estimate of OH concentration and calculate steady state HONO. I think this is important to show how the daytime HONO observed cannot be explained by standard chemistry.

*The resulting HONO PSS state concentration is similar in magnitude and diurnal shape to other literature examples , e.g. in London (Lee et al., 2016). A plot of the HONO PSS along with the average observed HONO concentration is now included in the manuscript (part of figure 9), and details of the HONO PSS calculation and result form an updated introduction to Section 3.5 "Possible daytime HONO sources".*

6. Some attempt should also be made to calculate the source of HONO from other postulated mechanisms (e.g. surface $NO_2$ conversion, soil based emission) to give some idea as to whether these mechanisms can produce the daytime HONO observed. Again, it is difficult to do this without some of the supporting data, however estimates could be made based on measurements in other cities in the literature.

*Michoud et al., 2014 and Lee et al., 2016 provide useful parameterisations for many different HONO sources including direct HONO emission from traffic, conversion of $HNO_3$, aerosol-mediated $NO_2$ to HONO conversion and ground-mediated $NO_2$ to HONO conversion.*

- *Calculating the expected aerosol-mediated conversion of $NO_2$ requires measurements of aerosol surface area (not available). In addition, as discussed at (previously) page 18, lines 1-6 the observed correlation is low between aerosols and HONO suggesting that aerosol-mediated processes cannot explain the observed HONO.*

- *Neither $HNO_3$ nor nitrate was measured at Broadmeadows and is very difficult to estimate without appropriate emission factors.*

- *Again, without appropriate emission factors for $NO_x$, calculating the HONO due to traffic is very difficult however the diurnal cycle of the HONO/$NO_2$ ratio, which is always > 1%, strongly suggests that traffic emissions cannot explain the observed HONO.*

- *Photoactivated daytime ground conversion of $NO_2$ is discussed in Lee et al., (2016), and the parameterization therein has been used to estimate the contribution of $NO_2$ ground conversion in Melbourne. The result now appears in Fig. 9 along with the calculated HONO PSS concentration demonstrating that the PSS + maximal $NO_2$ ground conversion rate in Lee et al., (2016) cannot explain the observed daytime HONO.*

- *Direct quantification of the soil emission contribution to the observed HONO is impossible without more information on local soil properties ? this is being followed up and will be the subject of further work. However, the contribution can be estimated from (previously) Fig. 13 which shows how the HONO diurnal cycle varies with soil moisture. Since original submission of the manuscript, soil water content percentage (% SWC) has been located through the Australian Bureau of Meteorology which has enabled this figure to be remade using bin values comparable with the literature, rather than the previous arbitrary rainfall index. Comparison along these lines and in response to Reviewer 2 have been added to the Discussion section. In an attempt to show that the missing HONO budget may be closed by soil based emissions, literature values for HONO and NO fluxes have been added to the HONO PSS rate calculation and plotted alongside the observed missing daytime HONO production rate. This now appears in figure 9.*

7. It would be useful to have a table of HONO and $NO_2$ levels from the literature from other cities round the world. Whilst there is some mention of comparisons in the text it would be clearer if this was brought together in tabular form to allow for easy comparison.

*To address this and a similar comment from Reviewer 2, a table of urban HONO and $NO_2$ measurements reported in the literature has been included in a Supplementary Information document and referenced in the manuscript.*

---

## Author Comment (AC2) · 9 Aug 2018

AUTHOR RESPONSES IN BLUE ITALIC TEXT

1. *General remarks: We wish to thank the referee very much for their interest in, and support of, the manuscript as evidenced by their very helpful comments. The suggestions for improved layout, extra references and improved linear regression analyses have greatly improved the flow and quality of analysis in the manuscript.*

2. There is a single major revision to this work that should be made regarding regressions. It is not clear what regression approach the Authors used, but atmospheric

datasets typically require accounting of error in both measures (e.g. Wu and Yu, 2018). The authors present a Pearson correlation coefficient in Figure 7 and discuss correlations heavily in their discussion from there on. This suggests that a linear least-squares analysis was used, which assumes error in the ordinate alone. The authors should clarify this and present appropriate regression metrics (e.g. slope and regression coefficient) when discussing correlations. Much of the discussion surrounding these comparisons is qualitative while the wording suggest that quantitative evaluations have been made. Including this quantitative information will strengthen the discussion.

*The Reviewer is correct in assuming that a linear least-squares method was used in the regression analyses presented, and as such errors in both measures were not appropriately accounted for. To address this, all linear regressions have been recalculated using the Deming regression method, described by e.g. Wu and Yu, 2018, which does allow for errors in both x- and y-variables. The Deming method includes a ratio of variances term (RV) which allows for the comparison of two methods with different inherent random analytical variability. In cases where two variables calculated from the MAX-DOAS retrieval are compared (for example retrieved HONO and $NO_2$ surface concentrations) RV has been assumed to be equal to unity. In cases where MAX-DOAS retrieved quantities are compared to external data sources (MAX-DOAS AOD vs MODIS satellite AOD, and MAX-DOAS $NO_2$ concentration vs EPA $NO_2$ in situ data), RV was calculated from the ratio of variances of each technique. Following from the increased confidence in the regression, we have tried to direct the discussion in a more quantitative manner as suggested by the Reviewer. Specific examples are highlighted in further response points below.*

3. Reference to figures throughout the manuscript should be capitalised as 'Fig. X' instead of 'fig. x'. *Corrected. All instances of "fig." have been replaced with "Fig."*

4. Page 3, Lines 30-35: Interferences from clouds, as presented in the discussion, should be included in the drawbacks here.

*Corrected. Updated text reads: "However, the MAX-DOAS method has some drawbacks which include complicated, multi-step data processing, limited information content from which to derive vertical profile information and interference from clouds.*

5. Page 4, line 30: repetition of 'and' to be corrected

*Corrected, thank you*

6. Page 5, line 11: differential slant column density is presented here and presumably is the source of the 'dSCD' term used later in the paper. Please define here, if this is correct.

*Updated. Text now includes "differential slant column density (dSCD)" for clarity.*

7. Page 6, figure 2: Keep the traces in panel c) consistent with the caption. Suggest switching $O_3$ and HCHO in the caption to be in the same order as the traces.

*Caption updated so that HCHO is listed before $O_3$ as in the figure traces and legend.*

8. Figure 2 and Table 1 could be moved to a supporting information document to reduce manuscript length

*Figure 2 and Table 1 moved to a Supplementary Information (SI) document as suggested.*

9. Figure 3 could be moved to a supporting information document to reduce manuscript length. Figure 4 could be easier to interpret if the time axis is consistent with the others in the manuscript. Two hour time intervals here, and in other diurnal plots, would provide the most detail without becoming cluttered. Figure 4 could benefit from being presented with larger panels if the other figures and table are moved to supporting documentation.

*The spectral detection of HONO, as shown in figure 3, underpins the results of the paper and therefore we would like to keep figure 3 in the main body of the text. Figure 4 has been updated to have two-hourly labels on the time-axis, and the panels have*

*been made larger to aid the reader.*

10. Page 8: Section 2.3 is quite long and is detailed for the manuscript, but feels like a lot of detail has also been left out. A suggestion here would be to simplify and condense this section further to improve its clarity (e.g. Equation 1 is not accessible to those not familiar with all of the literature in this section and could be considered superfluous along with many of the details) or some of the details could be moved to supporting documentation and expanded upon in the interests of allowing greater ease of reproduction of this detailed work.

*We agree that it is difficult to find the balance in Section 2.3 between accessibility and providing sufficient background mathematical detail to understand the retrieval methods used. In attempting to strike this balance we have re-worded text around equation 1, preferring to leave it in the manuscript as it provides the mathematical context to the discussion of the averaging kernels, degrees of freedom for signal and the retrieval parameter sensitivity tests in section 3.1. The readability of section 2.3 has been improved by shortening it considerably in response to this comment and point 12 below, shifting the latter half to the results section.*

*Previous text surrounding equation 1: "In order to retrieve trace gas vertical profiles in this way, information on the atmospheric aerosol extinction is needed to constrain the light path. This is determined using by applying the inversion algorithm to $O_4$ dSCD measurements. The solution for the aerosol profile **x** is determined iteratively with input aerosol properties being varied so as to minimise the cost function, given by $\chi^2$, i.e. the difference between the measurement vector y and the RTM simulations: ..."*

*Revised text surrounding equation 1: "In order to retrieve trace gas vertical profiles in this way, information on the atmospheric aerosol extinction is needed to constrain the light path. The solution for the aerosol profile **x** is calculated iteratively by varying the aerosol input parameters until the difference between the measurement vector **y** and the RTM simulations is minimised. This difference is given by the cost function $\chi^2$: ..."*

11. Page 8, Line 14: K presented as a weighting function matrix does not appear in either Equation 1 or Equation 2. Please revise.

*Corrected. The description of the weighting function matrix K applies to a different form of equation 1, which was included in a previous version of the manuscript. Reference to "K" has now been omitted.*

12. Page 9, Lines 9-23: These are results. Suggest relocating to the results and discussion section.

*To shorten Section 2.3 and in line with the Reviewer's suggestion, the profile retrieval results shown in (previously) figure 5, and its associated discussion (from page 8 line 10) have been shifted to the Results section. They now form part of Section 3.1 "Vertical distribution of aerosols, $NO_2$ and HONO".*

13. Page 10, Figure 5: There are acronyms (or short-hand notation) being used in the upper row of panels which are not defined in the caption. Please do so. In the bottom row of panels, there are 20 different lines presented in each panel and the values for each are very small in the legend. Are all of these necessary or can half of them be removed without undermining the findings? It would allow all panels in this figure to be increased in size and make it easier to read. Finally, panels (a) and (b) are not labelled here. Please add these.

*The averaging kernel labels at the bottom of (previously) figure 5 were indeed busy and have been updated for clarity. Acronyms and short hand notation in the profile figure legends have been replaced with clear and specific labels, the panels have been labelled a, b and c and the caption modified to describe the updated figure.*

14. Page 10, Line 4: repetition with 'the"

*Repetition removed, thank you*

15. Page 10, Lines 13-14: 'by a bias towards the a priori due to lower measurement sensitivity at these levels'. This explanation is unclear. Please consider revising to

improve the clarity here.

*Text previously read "The vertical profile reveals a dominant contribution of a priori shape parameters (scale height and ground extinction, blue colours) to upper level uncertainty, with a 60 % error contribution above 500 m. Below 500 m, the influence of the shape parameters is much less significant at 10 %, while the optical properties (yellow and green colours) play a more significant role with a 12 % error. The observed higher sensitivity of the retrieved profiles to the a priori at high altitudes can be explained by a bias towards the a priori due to lower measurement sensitivity at these levels."*

*This has been revised to: "The vertical profile reveals a dominant contribution of a priori shape parameters (scale height and ground extinction, blue colours) to upper level uncertainty, with a 60 % error contribution above 500 m. This is expected since the inherent higher sensitivity of the retrievals to the measurements at low altitudes, means the a priori profile more strongly constrains the retrieval at high altitudes. Below 500m, the influence of the shape parameters is much less significant at 10 %, while the optical properties (yellow and green colours) play a more significant role with a 12 % error."*

*As described in point 17 below, this text along with Fig. 6 has now been shifted to the SI document.*

16. Page 11, Figure 6: This figure could be moved to a supporting document. The panels are alphabetically labeled, so the text boxes for each can be removed. VCD is not defined in the caption. Also add a note regarding the exponent terms for $NO_2$ and HONO VCD values as they may be easily missed.

*Fig. 6 has been moved to the SI document and labels highlighting vertical column density (VCD) and noting the trace gas column exponent terms have been added to the caption. To further shorten the manuscript, we have decided to move the bulk of Section 3.1 to the SI document since the key finding of the sensitivity tests, namely confidence in the profile retrievals, is summarised by the error shading on the example*

*profiles in (previously) figure 5. This allows for a more concisely structured results section.*

17. Page 11, Line 10: 'the lowest 500m' should be 'in the lowest 500 m'.

*Correction made (text now in SI document).*

18. Page 11, Lines 11-14: Is the 'high sensitivity' instrumental sensitivity or retrieval sensitivity? Use of the word sensitivity here is a bit unclear and the retrieval sensitivity might be more intuitively termed 'potential error' or 'estimation error' or simply 'error'. What is a 'low error budget'? This wording is not consistent with the rest of this section. Please clarify.

*The 'high sensitivity' referred to is the sensitivity of the retrieval to the true profile. 'Low error budget' was perhaps a poorly worded way to indicate that the contribution of errors from a priori and forward model parameters was low. This result is explained in more meaningful detail in what is now Section 3.1. Therefore the revised sentence described below, appearing now in the SI document, has been simplified for greater clarity.*

*The sentence previously began "The high sensitivity to the ground level retrievals as demonstrated by the HONO and $NO_2$ averaging kernels, combined with the low error budget due to smoothing, noise, aerosol and shape a priori parameters in the lowest 500 m, gives confidence in the measured trace gas ground concentrations." This wordy and confusing sentence has been revised to: "The low retrieval errors and high sensitivity to the true atmospheric profile at low altitudes, as demonstrated by the averaging kernels and sensitivity tests presented in (previously) Fig. 5, gives confidence in the measured trace gas surface VMRs".*

19. Page 12, Figure 7: From here on forward, the regression analyses should be clearly presented. A useful quantity that would have been obtained in the analysis here is the slope, which gives some indication of the bias that is discussed qualitatively. Such

bias is expected given that the in-situ monitors are located at ground level (on top of sources), while the MAX-DOAS is observing more dilution of those sources. In the caption of this figure there is a reference to 'surface concentration' measured by the MAX-DOAS, but isn't the lowest elevation angle somewhere between 100 - 500 m above ground level? This isn't terribly clear and could help bridge these observations with ground observations more easily.

*We thank the reviewer for the constructive advice regarding the regression analyses. As noted above, the Deming method has now been applied to all regression analyses in the manuscript. This is introduced at (previously) section 3.2 when the MODIS and MAX-DOAS AOD measurements are compared.*

*Previous text: "Consistent with these limitations, while the ranges for MODIS, averaged over a 10 km spatial radius around Broadmeadows, and MAX-DOAS AOD were very similar (AOD varying between 0.05 and 0.2), the temporal correlation was weak at only 0.33. A longer sampling period and more local compatible datasets, such as PM2.5 measurements, are therefore needed for a useful validation of the MAX-DOAS aerosol 10 results."*

*Revised text: "Regression analysis was conducted using the Deming method which, unlike simple linear least squares regression, assumes measurement error in both x and y variables. It also allows for the regression to be weighted by the ratio of variances (RV) between the independent and dependent variables. In this case RV ($V_{\mathrm{MAXDOAS}}/V_{\mathrm{MODIS}}$) was 0.37 and the regression analysis showed a slope of 2.18 and Pearson's R coefficient of 0.33. Therefore, while the ranges for MODIS, averaged over a 10 km spatial radius around Broadmeadows, and MAX-DOAS AOD were very similar (AOD varying between 0.05 and 0.2), the MAX-DOAS AOD was typically half the MODIS-retrieved AOD. Addressing such discrepancies between ground and satellite-based retrievals are an important ongoing research area, with longer sampling periods and local compatible datasets such as ceilometer, LIDAR or PM$_{2.5}$ measurements needed for a confident validation of the MAX-DOAS aerosol results."*

*Further, the reviewer makes a good point regarding the use of the term "surface concentration". In fact, the retrieved quantity is the average mixing ratio over the lowest retrieval layer, which is 0-200 m. This is addressed at (previously) section 3.2 when comparing the EPA and MAX-DOAS $NO_2$ VMRs, which now also includes quantitative data from the Deming regression analysis.*

*Previous text at page 12, line 16-20: "Given the wide spatial range of the four EPA measurement sites, the possibility for widely varying local meteorological conditions at each site, and the fundamentally different measurement techniques, a correlation of 0.56 is a positive result for this comparison. When the local wind direction at Broadmeadows was from the south-west, correlation between the EPA stations directly to the south-west (Altona North and Footscray) the Broadmeadows MAX-DOAS $NO_2$ improved to 0.66, a positive 20 result which provides the strongest external validation available for these the MAX-DOAS trace gas retrievals."*

*Revised text: "The Deming regression analysis included an RV $(VAR_{MAXDOAS}/VAR_{EPA})$ of 1.25 and showed a Pearson's R coefficient of 0.58 and slope of 1.66. The slope of the linear regression highlights that the EPA values are typically higher which might be expected given that the EPA instruments measure in-situ ground level $NO_2$ while the MAX-DOAS "ground VMR" in fact samples the lowest 200 m of the troposphere through which the surface concentration is diluted. Furthermore, given the wide spatial range of the four EPA measurement sites and the possibility for widely varying local sources and meteorological conditions at each site, the correlation of 0.58 is a positive result for this comparison. When the local wind direction at Broadmeadows was from the south-west, correlation between the EPA stations directly to the south-west (Altona North and Footscray) the Broadmeadows MAX-DOAS $NO_2$ improved to 0.66 (although no change in regression slope was observed), a result which provides the strongest external validation available for these the MAX-DOAS trace gas retrievals."*

20. Page 12, Line 9: 2.5 should be a subscript. In addition, wouldn't ceiliometer or

LIDAR measurements be more useful in validating the MAX-DOAS aerosol results?

*Subscript added. Indeed, local ceilometer data and LIDAR data would be appropriate, these techniques have been added as suggested - revised text is included in the response to point 19*

21. Page 12, Line 17: The technique used by the EPA has not been presented and should be added as supporting instrumentation details in section 2.1. Presumably these are chemiluminescent analyzers with molybdenum converters?

*The Reviewer is correct in this assumption, a sentence noting the EPA $NO_x$ measurement technique has been added as suggested.*

22. Page 12, Line 19: What is the slope of the comparison? What type of regression was used?

*Corrected - addressed in the revision of the regression analyses in Section 3.2, in point 19.*

23. Page 13, Lines 6-9: Add appropriate quantitative regression data here, along with coefficient values that justify selection of wording such as 'correlated strongly'. The direction of the correlation is also important. Was the relationship a positive or negative one in each of these cases? Please provide these quantitative details.

*The key message here was to show that the colour index had some independent external validation, namely from the solar radiation measurements at Melbourne Airport. Deming regression analysis for the MAX-DOAS colour index vs the mean global irradiance, each parameter being normalised by their respective mean values, gave a slope of 1.75 and Pearson's R coefficient of 0.77, indicating that the values were correlated as expected. Further validation came from the fact that the slope became closer to 1:1 (1.65) and the Pearson's R coefficient increased (0.85) when the cloud filter was applied showing that periods of decreased colour index corresponded with periods of decreased global radiation nearby. Since this is the strongest external validation and*

*the discussion has been extended, reference to correlation between the HEIPRO retrieval errors and decreased colour index have been removed for clarity and brevity.*

24. Page 13, Lines 14-17: These ratios have been derived from measurements made at night or from tunnel studies and their applicability to the interpretation of daytime data is questionable. During the day, the longer lifetime of $NO_2$ relative to HONO could result in the observed diurnal pattern of HONO/ $NO_2$ by simple boundary layer mixing processes diluting the surface $NO_2$ while the surface HONO source does not change (i.e. it could be independent of $NO_2$, as suggested by the weekend dataset). It may be worthwhile to discuss this further and carry it into the later discussion or to remove HONO/ $NO_2$ as a suitable daytime metric entirely.

*In light of this important point, the discussion around the HONO/$NO_2$ ratio has been changed. While little can be inferred from the relationship relative humidity and HONO, $NO_2$ or the HONO/$NO_2$ ratio (see also discussion of this at point 36 below), there are daytime, ambient air observations to support a HONO/$NO_2$ ratio < 1 % being indicative of traffic direct HONO emissions (e.g. Elshorbany, 2009). Therefore, at Page 13, lines 14-17 the discussion will focus on the fact that the HONO/ $NO_2$ ratio allows us to expect a secondary chemical source of the observed HONO, rather than direct HONO emission from the adjacent road corridors. Furthermore, given that Fig. 8 has now been shifted to the SI document, the discussion now centres on the diurnal cycle plots rather than the timeseries plots.*

*Previous text: "The ratio HONO/$NO_2$ has been used previously to categorize emission sources of HONO with HONO/$NO_2$ < 0.01 indicating direct emission dominates HONO production, HONO/$NO_2$ 0.01 to 0.03 indicating $NO_2$ to HONO conversion at low relative 15 humidity and HONO/$NO_2$ > 0.03 indicating $NO_2$ to HONO conversion at high relative humidity (Wojtal et al., 2011; Hendrick et al., 2014; Qin et al., 2009). Figure 8 shows that periods of peak HONO/$NO_2$ are more commonly a function of low $NO_2$ than high HONO, and that high HONO corresponds typically to HONO/$NO_2$ around 0.03."*

*Revised text (with reference to the diurnal cycle plots now): "The HONO/$NO_2$ ratio has been used previously to categorize emission sources of HONO with HONO/$NO_2 < 0.01$ indicating direct emission dominates HONO production (Wojtal et al., 2011; Hendrick et al., 2014; Qin et al., 2009, Elshorbany:2009). Periods of peak HONO/$NO_2$ were found to be more commonly a function of low $NO_2$ than high HONO (see also timeseries in Fig. S3 in the Supporting Information), with high HONO typically corresponding to HONO/$NO_2$ around 0.03. Given that the HONO/$NO_2$ ratio is consistently greater than 0.01 it is inferred that the observed HONO cannot be attributed to direct traffic emissions from the adjacent road corridors."*

25. Page 13, Lines 33-35: Quantitative values for 'correlated strongly'. This is consistent with literature reports of surface processes dominating over aerosol $NO_2$ conversion. Please cite some examples of this.

*Upon checking the mixing layer height ($H_{ML}$) calculations, it was found that a mistake had been made, using an incorrect conversion factor. With this corrected, the correlation (using the Deming Method) is in fact not very strong, as originally stated, but rather weak with a Pearson's R coefficient of 0.42. The regression analysis shows a slope of 1.65 for HONO vs $NO_2$ indicating that $H_{ML}$ is typically higher for $NO_2$ than HONO. This fits with the second original statement that vertical column density and surface mixing ratio correlate more strongly for HONO than $NO_2$. Similarly, to point 39 below, these findings are both now consistent with previous reports of surface processes dominating HONO production (e.g. Michoud et al., 2014, Lee et al., 2016).*

*Previous text: "$H_{ML}$ values for $NO_2$ and HONO correlated very strongly and were consistently shallow at around 500-700 m on sunny days. However, the correlation of vertical column density with surface mixing ratio, which is 0.89 for HONO and 0.80 for $NO_2$, suggests that surface values are a greater influence on the total column for HONO than $NO_2$."*

*Revised text: "Regression analysis using the Deming method showed that $H_{ML}$ values*

*for HONO and $NO_2$ were weakly correlated (Pearson's R coefficient 0.42), with a slope of 1.65 indicating that $H_{ML}$ was typically higher for $NO_2$ than HONO. This is consistent with the correlation of vertical column density with surface mixing ratio, which is 0.89 for HONO and 0.79 for $NO_2$, suggesting that surface values are a greater influence on the total column for HONO than $NO_2$. These findings are consistent with previous findings that HONO production is dominated by surface processes rather than at higher altitudes, such as aerosol-mediated conversion of $NO_2$ (e.g. Michoud et al. (2014); Lee et al. (2016)).*

26. Page 14, Figure 8: This figure could be moved to the supporting information document. Remove text boxes on each panel. Label each alphabetically. If necessary, clarify what is on each panel in the caption. Reduce the number of labelled ticks on each ordinate axis.

*Figure 8 has been moved to the SI document and changes made to the figure labelling as suggested.*

27. Page 14, Figure 9: Change time axis to two-hour intervals. The date format on top of each column is different from that in Figure 8. Keep date formats consistent throughout C5 the manuscript and consistent with ACP guidelines. Remove 'conc.' from the HONO and $NO_2$ labels. They are correctly identified as mixing ratios in the caption.

*Date and time labels have been updated for consistency with other figures, as suggested, and 'conc.' has been removed from the trace gas labels.*

28. Page 15, Line 7: Delete 'well'

*Corrected, thank you.*

29. Page 15, Lines 27-28: These daytime values are higher than might be expected given that the measurement is being made through a large volume and from the $NO_2$ intercomparison. This would suggest in-situ HONO measurements might exceed 0.5

ppb. How does the > 0.2 ppb HONO value compare to other reports in urban and rural environments?

*To address this (and a comment of Reviewer 1) a table of HONO VMRs from urban areas around the world has been compiled and added to the SI document. It can be seen from this table that maximum ground-level HONO VMRs >0.2 ppb can be expected in urban areas. The unusual aspect of the Melbourne measurements is the timing, rather than the magnitude, of the peak. To draw the reader's attention to the table, the text at the start of Section 3.5 has been revised:*

*Previous text: "During the three month measurement period, 33 days which were mostly sunny had peak HONO concentrations greater than 0.2 ppb. These periods allow analysis of the diurnal cycles of HONO, $NO_2$ and aerosol extinction, which are shown in fig. 11."*

*Updated text: "During the three month measurement period, 33 days which were mostly sunny had peak HONO concentrations in the lowest retrieval layer greater than 0.2 ppb. From the measurement timeseries (see example timeseries in the Supplementary Information), characteristic ranges for retrieved surface were found to be 0 to 0.35 km-1 for aerosol extinction, 0 to 30 ppb for $NO_2$ and 0 to 0.5 ppb for HONO. These values for HONO lie within the range of observed VMRs in urban areas around the world (see Table S2 in the Supplementary Information)."*

*The question of these observations compared to rural measurements is addressed at point 33 below.*

30. Page 15, Lines 32-33: The authors discuss that HONO does not peak in the early morning during their daylight observations. Are there examples of MAX-DOAS observations capable of seeing the previous night's HONO prior to photolysis? What are the vertical resolution differences between these MAX-DOAS measurements and how might that impact the observations (i.e. if the 'surface' bin is deeper than other observations, you'd expect to observe lower levels). Also, are there limitations in the

MAX-DOAS measurement near sunrise with the instrumental orientation that could result increase the error in capturing a quantitative absorption signal for HONO at this time?

*To our knowledge the only previous papers reporting HONO vertical profiles from MAX-DOAS measurements were Hendrick et al., 2014,(Beijing) and Garcia-Nieto et al., 2018 (Madrid). The latter has been published in the intervening period between submission of this manuscript and the Reviewer responses so has now been duly included, with each being cited in several places throughout the manuscript. Hendrick et al present diurnal profiles of HONO VMR from the lowest retrieval layer, for the different seasons. The vertical resolution of the retrieval layers in this paper is the same as in Hendrick et al 2014. In each case the diurnal cycle maximum is at the start of the day, followed by a steady decrease in HONO VMR over the course of the day in line with HONO VMR diurnal cycles in other urban centres measured using different techniques. This suggests that the MAX-DOAS technique can reasonably be expected to detect pre-photolysis morning HONO if it is present.*

*Nevertheless, retrievals from MAX-DOAS data close to both sunrise and sunset are challenging because sunlight traverses its maximum pathlength through the atmosphere at these times. In the DOAS analysis this is accounted for by optimising fitting parameters (such as choice of DOAS polynomial) and using zenith reference spectra from the most recent set of elevation scans to ensure effective cancellation of stratospheric interference. However it was found that DOAS fit residuals were significantly higher (>1x10-3) for solar zenith angles greater than 80o compared to 4x10-4 for solar zenith angles in the middle of the day, which in turn led to larger differences between modelled and measured dSCDs in the profile retrieval. Hence results are reported for data <80o SZA. Therefore, results presented are missing the first 30 min of daylight in which, potentially, pre-photolysis HONO could exist. Despite this, given that all previous urban HONO diurnal cycles show a decrease across the whole morning, the lack of any morning HONO observed in Melbourne is still considered a significant result.*
*To clarify this important point the following sentence has been added to (previously) section 3.5: "It should be noted that due to increased DOAS fit residuals and consequent profile retrieval errors for solar zenith angles (SZA) greater than 80o, no data from SZA > 80o is presented. During autumn in Melbourne this means corresponds approximately 30 mins after sunrise and 30 min before sunset."*

31. Page 16, Figure 10: Consider moving this figure to a supporting information document. The discussion does a good job of conveying the information presented here.

*We believe that Fig. 10 provides an important spatial overview of potential trace gas sources and would like to keep it in the main text.*

32. Page 16, Lines 1-5: It would improve the discussion to report the daytime maximum mixing ratios observed in these other locations, for context.

*Text previously read: "Previously such daytime maxima in the HONO diurnal cycle have only been observed in rural locations for example at a rural site in Germany (Acker et al., 2006), a forested site in Michigan USA (Zhou et al., 2011), and in rural Cyprus (Meusel et al., 2016) although in each case the peak diurnal HONO value averaged significantly less than observed in Melbourne."*

*Revised text reads: "Previously such daytime maxima in the HONO diurnal cycle have only been observed in rural locations for example at a rural site in Germany (Acker et al., 2006), a forested site in Michigan USA (Zhou et al., 2011), and in rural Cyprus (Meusel et al., 2016). In each case however, the maximum HONO VMR observed was less than in Melbourne, at 110 ppt, 70 pptv and 100 pptv respectively."*

33. Page 17, Figure 11 (and other similar instances): The caption does not describe the panels correctly here. Further, the caption description for the similar panels can be improved by changing the phrase directed for the first panel to the following: 'Diurnal cycle plots for the 1 hourly averages of (a) $NO_2$, (b) HONO, (c) $HONO/NO_2$, and (d) aerosol extinction surface values at Broadmeadows'. The alphabetical indicators for

each panel should be displayed outside the axes throughout this figure.

*The caption has been updated and the alphabetical labels placed outside the axes as suggested. (Previously) Fig. 12 has also been updated to include (a) and (b) which were missing.*

34. Page 18, Line 7: The citations here are not in the proper format to present via 'e.g.'. This is also a hanging sentence. Please correct it.

*Corrected. Text previously read: "The HONO diurnal profile observed in this campaign matches temporally with the diurnal profile of the missing HONO production source calculated in both rural e.g. (Meusel et al., 2016) and urban e.g. (Wong et al., 2012; Pusede et al., 2015)."*

*Revised sentence: "The HONO diurnal profile observed in this campaign matches temporally with the diurnal profile of the missing HONO pro- duction source calculated in both rural (e.g., Meusel et al. (2016)) and urban areas (e.g., Wong et al. (2012); Pusede et al. (2015))."*

35. Page 18, Lines 11-18: These ratio values continue to be potentially misleading. Suggest careful revision or even removing this part of the discussion since intensive chemical description of the HONO/$NO_2$ ratio under sunlit conditions has not been well established.

*The reviewer makes a sound point that the discussion of these ratios in sunlit conditions has not been well established. In attempting to understand any relative humidity dependence of the HONO results, the regression analysis presented in this part of the discussion, and the lack of distinct relative humidity trends in the HONO vs $NO_2$ plot (previously Fig. 12) are stronger evidence. Therefore, discussion of the relative humidity dependent HONO/$NO_2$ ratios has been removed from this part of the discussion.*

*Text previously read: "The heterogeneous conversion of $NO_2$ on wet surfaces accord-*

*ing to reaction R5 has been suggested as a primary HONO source pathway (Wong et al., 2012) especially during the night when there are no OH radicals available to form HONO via reaction R4. Values of the HONO/ $NO_2$ ratio between 0.01 and 0.03 have been found when the conversion via reaction R5 proceeds in low relative humidity environments while HONO/ $NO_2$ > 0.03 indicates 15 conversion at high relative humidity. The average midday HONO/ $NO_2$ ratio averages around 0.035 (fig. 11(c)) in this case, however fig. 12(b) shows that most of the data points where HONO/ $NO_2$ is between 0.025 and 0.05 correspond to low relative humidity. The overall correlation between HONO surface concentrations and relative humidity is weak and negative at -0.31 further indicating that reaction R5 cannot explain the high daytime HONO."*

*Text now reads: "The heterogeneous conversion of $NO_2$ on wet surfaces according to reaction R5 has been suggested as a primary HONO source pathway (Wong et al., 2012) especially during the night when there are no OH radicals available to form HONO via reaction R4. However, while fig. 12(b) shows that most of the data points for HONO/ $NO_2$ > 0.025 correspond to relative humidity less than 50 %, there is no clear trend for HONO/$NO_2$ < 0.025. Regression analysis showed that the overall correlation between relative humidity and HONO VMR was very weak (slope -0.001, coefficient -0.162) further indicating that reaction R5 cannot explain the high daytime HONO."*

36. Page 18, Line 20: 'suggests that $NO_2$ is implicated in some other way'. The weekend data presented here suggests that this may not be true OR that the mechanism is $NO_2$-saturated. See some discussion of this in (Pusede et al., 2015).

*The opening sentence of this paragraph was intended as an introduction to the ensuing discussion of potential $NO_2$-based HONO sources, rather than a conclusion. To minimise the chance for confusion in relation to this, the paragraph opening "$NO_2$ is implicated" omitted. The weekend/weekday data is addressed again in this discussion, (previously) page 19 lines 8-13. The good point about $NO_2$ mechanisms potentially being saturated has been included in the discussion at (previously) lines 8-13 as follows.*

*Text previously read: "The correlation (between HONO and* $NO_2$*) holds as a function of wind speed, with both HONO and* $NO_2$ *being localised (fig. 10) although HONO is more dependent on wind direction. As discussed in the source distribution section above, the correlation does not hold between weekends and weekdays and these combined factors suggest that while plausible photo-activated, ground based* $NO_2$ *conversion mechanisms exist, the correlation does not necessarily entail high* $NO_2$ *to HONO conversion."*

*Revised text: "While the correlation (between HONO and* $NO_2$*) holds as a function of wind speed, HONO is more dependent on wind direction (see Fig. 10) and as discussed above, the correlation does not hold between weekends and weekdays. This suggests that while plausible photo-activated, ground based* $NO_2$ *conversion mechanisms exist, such mechanisms may be saturated and or of insufficient strength to account for the observed daytime HONO."*

37. Page 19, Figure 13: The data presented here suggest that there is a suppression of HONO daytime surface flux due to increased soil water content. There are a few instances of this hypothesis being tested under laboratory and field conditions that may be worth mentioning here (Donaldson et al., 2013, Donaldson et al., 2014;Oswald et al., 2013;Scharko et al., 2015;Su et al., 2011;Weber et al., 2015). Comparison to the microbial pathways, reversible partitioning, and surface adsorption/dissolution could all enhance the discussion.

*Many thanks to the reviewer for these helpful references which show that HONO and NO emissions are indeed expected for dry rather than moist soil. Following this point, along with point 39 and the comments of Reviewer 1, some considerable work has been put in to calculate in greater detail the 'missing HONO budget' in Melbourne. As part of this, modelled soil moisture content data has been obtained from the Australian Bureau of Meteorology which has allowed (previously) Fig. 13 to be remade with bin values comparable to, e.g., Oswald et al., 2013. Furthermore, literature values for potential HONO and NO soil fluxes have been plotted alongside the unknown HONO*

*production rate to show that soil emissions can plausibly close the local HONO budget. Reference to some the papers suggested above by the Reviewer has been made in this discussion. However given the current length and scope of the manuscript we would like to leave the more focused discussion of possible soil sources, including microbial pathways and partitioning, to a future study containing soil property measurements.*

38. Page 19, Lines 1-2: The literature has been clear on the aerosol surface area conversion of $NO_2$ to HONO being a minor daytime production route for some time. Suggest including some references to the literature that have demonstrated the phenomenon here in support of your findings.

*Yes, this has been found before for example in Michoud, et al., 2014 and Lee et al., 2016, citation of these works is now included. Following from the discussion at page 19 lines 1-2, text previously read "indicating that aerosol-mediated $NO_2$ conversion cannot explain the observed high daytime HONO levels."*

*Revised text: "indicating that aerosol-mediated $NO_2$ conversion cannot explain the observed high daytime HONO levels. This is in line with previous findings in, e.g., Michoud et al. (2014) and Lee et al. (2016)."*

39. Page 20, Line 17: This instance of biocrust discussion should be expanded if there are local biocrusts near the observation site, and generally throughout the Melbourne area. The established literature on this, coupled to anything known about regional biocrust microbial composition, may facilitate a stronger capacity to speak on this potential daytime HONO source instead of speculating.

*The reviewer is correct in suggesting that the previous discussion around biocrusts was speculative. Indeed, a more detailed description of potential biocrust instances and general soil characteristics in the Melbourne area is being followed up, and will be addressed in future work when more measurements are available. However, the discussion of potential soil-based HONO emissions has been tightened considerably in the re-written section as described at point 37 above and in response to Reviewer*

*1's extensive suggestions on source term calculations.*